# PLANNING WITH UNIFIED MULTIMODAL MODELS

## ABSTRACT

With the powerful reasoning capabilities of large language models (LLMs) and vision-language models (VLMs), many recent works have explored using them for decision-making. However, most of these approaches rely solely on language-based reasoning, which limits their ability to reason and make informed decisions. Recently, a promising new direction has emerged with unified multimodal models (UMMs), which support both multimodal inputs and outputs. We believe such models have greater potential for decision-making by enabling reasoning through generated visual content. To this end, we propose *Uni-Plan*, a planning framework built on UMMs. Within this framework, a single model simultaneously serves as the policy, dynamics model, and value function. In addition, to avoid hallucinations in dynamics predictions, we present a novel approach *self-discriminated filtering*, where the generative model serves as a self-discriminator to filter out invalid dynamics predictions. Experiments on long-horizon planning tasks show that Uni-Plan substantially improves success rates compared to VLM-based methods, while also showing strong data scalability, requiring no expert demonstrations and achieving better performance under the same training-data size. This work lays a foundation for future research in reasoning and decision-making with UMMs.

## 1 INTRODUCTION

Large language models (LLMs) and vision-language models (VLMs) have demonstrated strong reasoning capabilities across a wide range of tasks (Brown et al., 2020; Wei et al., 2022; OpenAI, 2023; Zhang et al., 2024b). Motivated by this, many recent works (Huang et al., 2022a; Ichter et al., 2022; Driess et al., 2023; Hu et al., 2023) have explored their application to decision-making, such as generating high-level, step-by-step plans for long-horizon tasks. However, their planning process remains purely text-based. Even for most VLMs, visual inputs are typically used only at the initial stage of reasoning, rather than being incorporated throughout the thinking process. As a result, the reliance on a single modality limits the model's ability to accurately represent the current state during planning. The absence of multimodality throughout the thinking process limits their effectiveness in complex scenarios that require accurate spatial or visual understanding.

Recently, an increasing number of works (Hu et al., 2024; Zhou et al., 2024; Li et al., 2025; Chern et al., 2025) have proposed incorporating images into the reasoning process. This is typically achieved by integrating external tools to interpret visual observations, for example, by generating program code to call Python plotting libraries (Hu et al., 2024), or by invoking vision models for segmentation or object detection on input images (Zhou et al., 2024). However, such approaches are highly dependent on separate visual modules or external toolchains, which limits their adaptability to more complex visual reasoning tasks. In contrast, a different line of work (Li et al., 2025; Chern et al., 2025) explores generating intermediate images directly within the model to support reasoning. Although this approach is more general and holds greater promise, it has been used primarily to visualize reasoning traces (Li et al., 2025) or to iteratively refine image generation (Chern et al., 2025), rather than to enable more sophisticated decision-making.

Notably, a promising new class of models has recently emerged, i.e., unified multimodal models (UMMs) (Wu et al., 2024; Xie et al., 2025b; Wang et al., 2024; Liao et al., 2025; Deng et al., 2025), which support both multimodal inputs and outputs, typically in the form of images and text. We argue that such models are particularly well-suited for decision-making, as they can simultaneously serve as dynamics models (generating the next visual observation), as policies (generating text-based

Figure 1: Overview of Uni-Plan, a planning system where one single UMM serves as (i) the policy, (ii) the dynamics model, (iii) the self-discriminator, and (iv) the value function simultaneously.

actions), and as value functions (estimating how far away from goals), thus enabling integrated planning. However, such models remain subject to the curse of the horizon, with a key bottleneck being their limited ability to serve as faithful dynamics models. While current state-of-the-art models can perform basic image-editing tasks (Deng et al., 2025; Liao et al., 2025), our findings indicate that they are still insufficiently accurate to give reliable dynamics predictions. This limitation can be alleviated through finetuning for relatively simple downstream tasks, but the improvement does not extend to more challenging scenarios, particularly those involving stochastic dynamics.

To address this challenge, we strategically leverage the UMM's flexible input–output modality to employ it as a self-discriminator for filtering out invalid transition predictions. Concretely, the model first generates multiple candidate predictions for the next observation. It then operates in an inverse dynamics mode, inferring the action that would lead from each current–next observation pair. By comparing these inferred actions with the actual action, we can identify and discard those transitions where the actions do not match, effectively removing implausible predictions. Building on this capability, we develop a planning framework *Uni-Plan*, and demonstrate its superior performance across a range of long-horizon planning tasks.

We highlight the main contributions of our work below:

- We propose *self-discriminated filtering*, where the generative model serves as a self-discriminator to filter out invalid dynamics predictions for a more accurate dynamics model.

- We present a planning framework *Uni-Plan*, illustrated in Figure 1, where one UMM plays the roles of (i) policy, (ii) dynamics model, (iii) self-discriminator, and (iv) value function simultaneously.

- Evaluating on several long-horizon planning tasks, our method achieves nearly 30% higher success rates than open-source VLM-based planning methods, and even matches the powerful GPT-5-Thinking model. Furthermore, our method also exhibits strong data scalability, requiring no expert demonstrations for finetuning and outperforming VLMs when trained with the same amount of data.

## 2 PLANNING WITH UNIFIED MULTIMODAL MODELS

### 2.1 FORMULATION

In this work, we focus on leveraging unified multimodal models (UMMs) for decision-making. Here, the UMMs refer to such models capable of multimodal inputs and outputs, typically in the form of images and text. This kind of model can give us more flexibility and higher potential when using them for decision-making. In this paper, we use BAGEL (Deng et al., 2025), the state-of-the-art open-source UMM, as the foundation model. We refer readers to Appendix A for more details.

We formulate the decision-making process as a hierarchical framework. At the high level, given an initial visual observation $o_0$ of the environment and a goal $g$ described in natural language, A model (VLM/UMM) is required to generate a sequence of plans $a_{0:H}$ to achieve the goal, where each $a_i$ is a textual action. At the low level, we assume the availability of a set of off-the-shelf policies (skills) $\{\pi^i\}_{1:N}$ that serve as controllers, producing low-level control actions conditioned

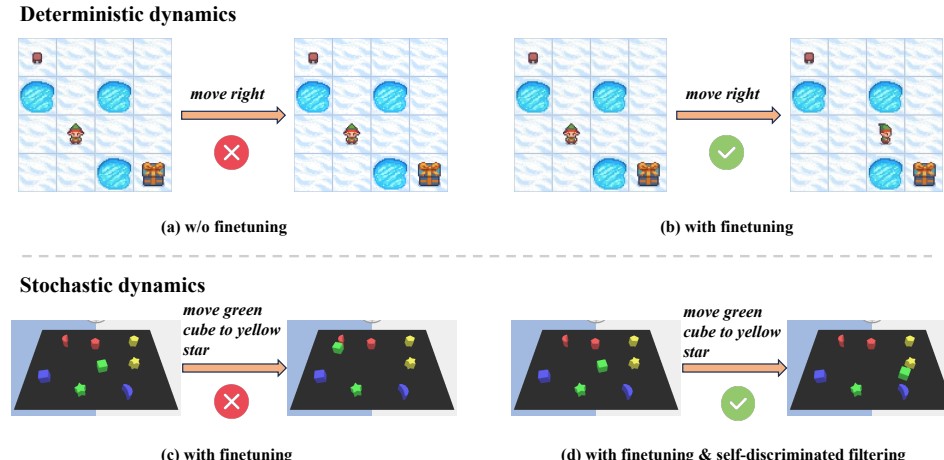

Figure 2: Illustration of dynamics predictions by different models.

on the current observation and the corresponding textual action. These low-level policies can be derived from either behavior cloning or reinforcement learning. Our work primarily focuses on the high-level planning component.

The core idea of our method is to employ a UMM for planning, which inherently integrates the functions of a dynamics model, a policy, and a value function. Owing to the flexible input-output modalities of UMMs, a single model can simultaneously serve all these roles. In Section 2.2, we first describe how to utilize it as a reliable dynamics model, which constitutes the most challenging aspect of the planning system. Subsequently, in Section 2.3, we present the design of the overall planning framework.

## 2.2 UMMs AS DYNAMICS MODELS

To enable high-level planning, the model must be capable of predicting the next visual observation conditioned on the current observation and a textual action, i.e., $P_{\mathrm{UMM}}(o_h, a_h) \rightarrow o_{h+1}$. This task closely resembles image editing in the training of UMMs (Deng et al., 2025; Liao et al., 2025), as both require accurate language grounding while preserving the consistency of unaffected regions in the image. However, dynamics prediction presents a greater challenge, as it additionally demands the ability to reason about precise causal effects of textual actions.

As an illustrative example, we employ BAGEL (Deng et al., 2025) to perform dynamics predictions in a maze environment (Figure 2(a)). Note that we directly use its open-source weights without any finetuning. While the model preserves overall image consistency, it fails to predict the character's position accurately. This limitation can, however, be mitigated through few-shot finetuning. As shown in Figure 2(b), after such adaptation, the model can serve as an effective dynamics model.

However, we find that only finetuning remains insufficient for accurate dynamics predictions on more challenging tasks, particularly those involving stochastic dynamics. In Figure 2(c), we illustrate the dynamics prediction for table rearrangement tasks using the finetuned BAGEL. The stochasticity in this setting arises from the existence of multiple valid outcomes $\{o_{h+1}^i\}$ for a given $(o_h, a_h)$. For example, suppose that the text action is "*move green cube to yellow star*", such that multiple valid next states may exist because the green cube could be placed at different relative positions around the yellow star, leading to different but equally correct results.

To mitigate this issue, we propose a novel technique termed *self-discriminated filtering*, which enables the model itself to act as a discriminator to select correct predictions from multiple candidates sampled by the model. Specifically, we jointly train a UMM to perform *inverse dynamics inference*, allowing it to predict the textual action that describes the transition between two observations, i.e., $P_{\mathrm{UMM}}^{-1}(o_h, o_{h+1}) \rightarrow a_h$. To identify valid transitions among the model's predictions $\{\hat{o}_{h+1}^i\}$ for $(o_h, a_h)$, we feed each candidate pair $(o_h, \hat{o}_{h+1}^i)$ into the model to obtain an inferred $a_h^i$, and then verify whether $a_h^i$ matches the ground-truth action $a_h$[1]. Although effective, we observe that the

---

[1]Exact lexical match is not required and semantic equivalence is sufficient. We can use regular-expression matching to extract environment changes for comparison.

---

**Algorithm 1** Uni-Plan

---

**Require:** Initial visual observation $o_0$, Language goal $g$

1: **Functions:** Dynamics model $P_{\text{UMM}}(o_h, a_h)$, Inverse dynamics $P_{\text{UMM}}^{-1}(o_h, o_{h+1})$, Policy $\pi_{\text{UMM}}(o_h, g)$, Heuristic value function $H_{\text{UMM}}(o_h, g)$

2: **Hyperparameters:** Action-branching factor $A$, Dynamics-branching factor $D$, Planning beams $B$, Max planning horizon $H$

3: plans $\leftarrow \left[ (o_0, []) \, \forall i \in \{0, \ldots, B-1\} \right]$            ▷ Initialize $B$ different plan beams

4: **for** $h = 0 : H$ **do**

5:     candidates $\leftarrow []$

6:     **for** $b = 0 : B$ **do**

7:        $o_h, a_{0:h} \leftarrow$ plans$[b]$            ▷ Get the observation and the action sequence

8:        $\{a_h^i\}_{0:A} \leftarrow \pi_{\text{UMM}}(o_h, g)$            ▷ Generate $A$ text actions

9:        **for** $i = 0 : A$ **do**

10:           **for** $j = 0 : D$ **do**

11:              $\hat{o}_{h+1}^j \leftarrow P_{\text{UMM}}(o_h, a_h^i)$            ▷ Generate a next observation

12:              **if** $P_{\text{UMM}}^{-1}(o_h, \hat{o}_{h+1}^j) = a_h^i$ **then**

13:                 candidates.append$((\hat{o}_{h+1}^j, a_{0:h} + a_h^i))$      ▷ Only add valid transitions

14:              **end if**

15:           **end for**

16:        **end for**

17:     **end for**

18:     plans $\leftarrow$ sort(candidates, $H_{\text{UMM}}$)$[0 : B]$            ▷ Keep the top-B beams

19:     **if** $\text{Max}(H_{\text{UMM}}) = 0$ **then**

20:        break            ▷ Early stop when number of steps left is 0

21:     **end if**

22: **end for**

23: plan $\leftarrow \arg\max($plans, $H_{\text{UMM}})$            ▷ Return highest value plan

---

model occasionally produces predictions in which objects are unexpectedly missing or duplicated. To address such rare errors, we further employ an object-count consistency check: the model counts the objects in both the current observation and each predicted next observation, and discards predictions where the counts differ, thereby reducing such anomalies. With this technique, we achieve a significant reduction in hallucinations, as shown in Figure 2(d). A quantitative evaluation of the improvements introduced by this approach is presented in Section 3.2.

## 2.3 WHOLE PLANNING SYSTEM DESIGN

In this section, we describe how to construct the complete planning system based on UMMs. To enable planning, two additional components are required: a policy for sampling actions, and a value function for evaluating states caused by different actions.

For the policy, we directly employ a UMM as a function $\pi_{\text{UMM}}(o, g) \rightarrow a$ that outputs a textual action given an observation $o$ and a goal description $g$. We do not finetune the model to serve as a policy. Instead, we provide it with task-specific instructions and constraints (i.e., the set of valid actions) through the prompt. During planning, this policy takes the current observation as input and samples multiple action candidates for evaluation. For the value estimation, we follow Du et al. (2024) to implement a heuristic function $H_{\text{UMM}}(o, g) \rightarrow u$, which takes as input an image observation $o$ and a goal $g$, and outputs a scalar estimating the number of steps required to reach a state that satisfies $g$ from the current state $o$. To construct this heuristic, we finetune the UMM on a labeled dataset in which each image observation is annotated with the remaining number of steps to the goal.

With all modules in place, we can now introduce the full planning system *Uni-Plan*. We adopt beam search as the underlying planning algorithm. First, beam search initializes $B$ parallel beams. Then, at each step of the planning horizon, for each beam, the policy $\pi_{\text{UMM}}(o, g)$ samples $A$ candidate actions. For each action, the dynamics model $P_{\text{UMM}}(o, a)$ generates $D$ possible next observations. Subsequently, the *self-discriminated filtering* module selects a valid prediction among these $D$ can-

didates and extends the corresponding beam with the chosen transition. After each rollout step, the heuristic value function $H_{\mathrm{UMM}}(o, g)$ assigns a score to each beam, and only the top-$B$ beams are retained. The final plan is determined by the beam with the highest heuristic value at the end of the planning horizon. The overall procedure is summarized in Algorithm 1.

It is noteworthy that, unlike existing approaches that finetune VLMs for decision-making (Driess et al., 2023; Mu et al., 2023), our framework does not rely on expert demonstration datasets. The forward and inverse dynamics models can be finetuned on any available transition data (expert or non-expert). The policy component requires no finetuning, while the heuristic value function only necessitates some labeled data to learn to estimate the number of steps to a goal.

## 3 EXPERIMENTS

In this section, we empirically validate three claims:

- Compared to VLM-based planning methods, our method is better at long-horizon decision-making tasks, including both navigation and manipulation tasks.
- The strong decision-making ability is rooted in the fact that the fine-tuned Unified Multimodal Model (UMM) serves as a highly generalizable dynamics model, and is further strengthened by our proposed *self-discriminated filtering*, which rejects implausible transitions to improve prediction accuracy.
- Our approach demonstrates superior data scalability in two aspects: it requires no expert demonstrations for finetuning and achieves stronger performance than VLMs when trained with the same amount of data.

### 3.1 EVALUATION OF PLANNING ABILITY UNDER OOD ENVIRONMENTS

**Tasks.** To comprehensively evaluate the planning ability of different models, we design experiments across three simulated environments and one real-world environment. (i) *FrozenLake*: a maze-like environment where the agent must plan a path from a start location to a goal while avoiding traps. This task primarily assesses the model's ability to perform long-horizon reasoning under strict safety constraints. (ii) *Mini-BEHAVIOR*: a series of grid-world embodied AI tasks in which the agent is required to navigate and complete specified goals, such as picking up a target object and placing it in another goal position. This environment emphasizes both navigation and goal-directed decision-making. (iii) *Language Table*: a tabletop object rearrangement task where the agent manipulates objects on the table to achieve a desired configuration shown in a target image. This setting evaluates the model's capacity for grounded language understanding and spatial reasoning in manipulation tasks. (iv) *Real World*: a more challenging real-world object-rearrangement scenario where the agent must precisely identify previously unseen objects and generate a coherent multi-step plan toward the goal.

**Training & Test Sets.** To evaluate planning under out-of-distribution (OOD) conditions, we construct training and test configurations that explicitly induce distribution shifts in each environment. (i) *FrozenLake*: the test set contains unseen layouts that differ from the training set in both map size and trap distribution. (ii) *Mini-BEHAVIOR*: the test set comprises novel maps with different start, object, and goal positions. (iii) *Language Table*: the test set uses distinct block configurations for both the initial and goal states. (iv) *Real World*: the test set uses distinct configurations with unseen objects and containers for both the initial and goal states. We collect only 500 trajectories for each task for finetuning, except for the real-robot task, where data collection was more costly and resulted in 200 trajectories. Notably, the datasets used to finetune the VLM baselines are expert demonstrations, whereas our approach requires no expert data. Instead, we collect an equal amount of non-expert trajectories to ensure a fair comparison. Dataset statistics and collection procedures are provided in Appendix B.1.

**Baselines.** We compare our approach against a range of planning methods based on vision–language models (VLMs).

For open-source baselines, we include the mainstream VLM Qwen2.5-VL (Bai et al., 2025) and additionally compare BAGEL-VLM, which is BAGEL restricted to its VLM-only mode and thus

Table 1: Success rates of planning with different methods.

| Model | Frozen Lake | Mini-BEHAVIOR | Language Table | Real World | Average |
|---|---|---|---|---|---|
| **Closed-Source** | | | | | |
| GPT-5 | 0.08 | 0.04 | 0.00 | 0.10 | 0.06 |
| GPT-5-Thinking | 0.44 | 0.30 | 0.82 | **0.80** | 0.59 |
| GPT-5-Thinking-Tool | **0.98** | 0.68 | **0.90** | **0.80** | **0.84** |
| **Open-Source (three training runs)** | | | | | |
| Qwen2.5-VL-7B-Ins | 0.33±0.04 | 0.00±0.00 | 0.14±0.03 | 0.07±0.02 | 0.14±0.02 |
| Qwen2.5-VL-7B-Ins-CoT | 0.37±0.02 | 0.15±0.02 | 0.36±0.01 | 0.22±0.02 | 0.28±0.02 |
| Qwen2.5-VL-32B-Ins | 0.43±0.05 | 0.07±0.03 | 0.28±0.06 | 0.18±0.06 | 0.24±0.05 |
| Qwen2.5-VL-32B-Ins-CoT | 0.49±0.02 | 0.51±0.01 | 0.57±0.02 | 0.33±0.02 | 0.48±0.02 |
| BAGEL-VLM (baseline, 7B) | 0.38±0.05 | 0.01±0.02 | 0.23±0.02 | 0.09±0.04 | 0.18±0.03 |
| BAGEL-VLM-CoT (baseline, 7B) | 0.43±0.02 | 0.48±0.00 | 0.51±0.02 | 0.26±0.02 | 0.42±0.02 |
| **Ours** | | | | | |
| Uni-Plan (14B-A7B) | 0.95±0.01 | **0.83±0.01** | 0.73±0.02 | 0.63±0.02 | 0.78±0.02 |

serving as a natural ablation of our method. For these models, we consider both *non-CoT* and *CoT* variants:

- **Non-CoT version**: The model is fine-tuned using only the final answers so that it outputs a complete plan directly.
- **CoT version**: The model is additionally provided with rationales (Wei et al., 2022) during fine-tuning, enabling it to think step-by-step at inference.

For closed-source VLMs[1], we choose GPT-5 (OpenAI, 2025) and its variants:

- **GPT-5**: A standard chat model with relatively limited reasoning ability. We prompt it to think step by step to produce a plan.
- **GPT-5-Thinking**: A stronger model trained to reason through reinforcement learning.
- **GPT-5-Thinking-Tool**: Extends GPT-5-Thinking with a code interpreter, allowing it to write and execute code for better problem solving.

Appendix B.3 lists the detailed prompts and CoT demonstrations for all these tasks.

**Main Results.** Table 1 reports the planning success rates of all evaluated methods, measured by invoking the corresponding low-level policies to execute the generated plans in the environments. Our approach consistently outperforms open-source VLM baselines by a substantial margin. In particular, compared with the BAGEL-VLM, our method achieves more than 60% higher success rates than its non-CoT variant and nearly 40% higher than its CoT variant across all tasks. This directly highlights the superiority of our planning system over traditional chain-of-thought (CoT) reasoning based solely on text, as both approaches share the same underlying model, yet ours more effectively exploits BAGEL's capabilities. We provide a detailed analysis of VLM-based planning failure cases in the Appendix C.2. When compared with the advanced closed-source model, our method still exhibits comparable performance, which is a promising result given that GPT-5-Thinking-Tool is a significantly larger model and possesses broader knowledge. This also indicates that, beyond the use of external tools to enhance a model's visual reasoning capability, leveraging the UMM's inherent multimodal generation ability can likewise significantly strengthen reasoning performance, providing a new perspective for improving visual reasoning in the future.

**Planning Visualizations.** We further showcase some planning examples produced by Uni-Plan on the real-world task. Notably, this scenario contains previously unseen objects and containers, making it particularly challenging: the policy must accurately recognize new items, and the dynamics model must predict correct transitions under novel visual conditions. As shown in Figure 3, our planning system generates coherent plans that are both plausible in their action sequences and consistent in dynamics predictions. Additionally, we test Uni-Plan on a more challenging scenario with an unseen background. It can be seen that the change in background does not bring any influence to the planning, indicating the great generalization ability of our method. More qualitative visualizations are provided in Appendix C.1.

---

[1]Because the closed-source models cannot be finetuned directly, we employ few-shot prompting for in-context learning instead.

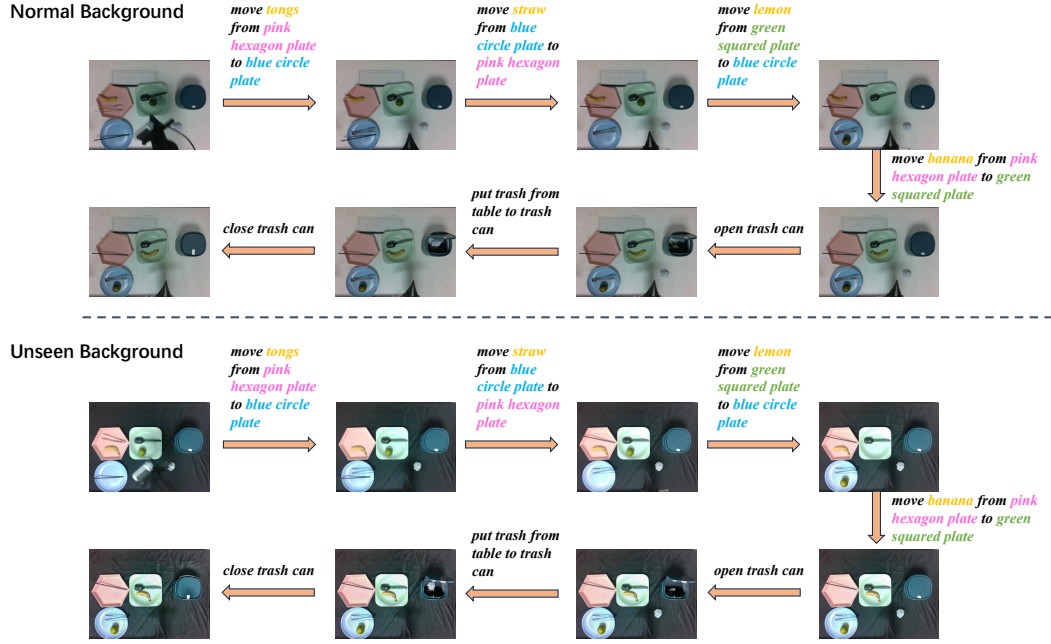

Figure 3: Planning for a real-world tabletop task with unseen objects and containers (top), and a more challenging case with additional unseen background (bottom).

## 3.2 STRONG DYNAMICS MODEL AS THE CORE OF STRONG DECISION-MAKING ABILITY

Unlike chain-of-thought (CoT) reasoning, which instinctively generates a reasoning trace in an autoregressive manner, our approach performs beam search over several sampled action candidates, allowing it to escape the limitations of a fixed policy and adapt to novel situations. However, this advantage hinges on the model's ability to accurately predict the outcomes of different actions.

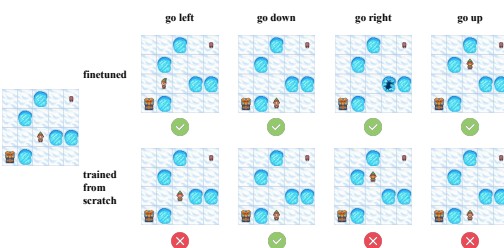

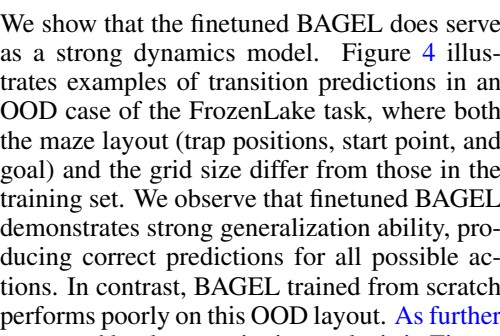

Figure 4: Illustrations of predictions on an OOD case by finetuned BAGEL and BAGEL trained from scratch as dynamics models.

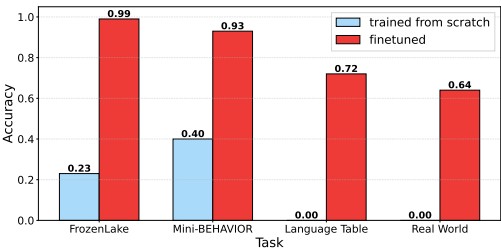

Figure 5: Quantitative comparison between finetuned BAGEL and BAGEL trained from scratch as dynamics models.

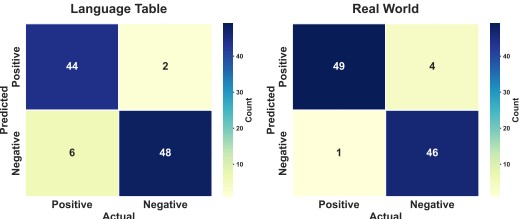

Figure 6: Confusion matrices of predictions by self-discriminated filtering.

We show that the finetuned BAGEL does serve as a strong dynamics model. Figure 4 illustrates examples of transition predictions in an OOD case of the FrozenLake task, where both the maze layout (trap positions, start point, and goal) and the grid size differ from those in the training set. We observe that finetuned BAGEL demonstrates strong generalization ability, producing correct predictions for all possible actions. In contrast, BAGEL trained from scratch performs poorly on this OOD layout. As further supported by the quantitative analysis in Figure 5, where we evaluate models on 100 OOD scenarios per task by measuring the accuracy of transition predictions, the finetuned model achieves substan-

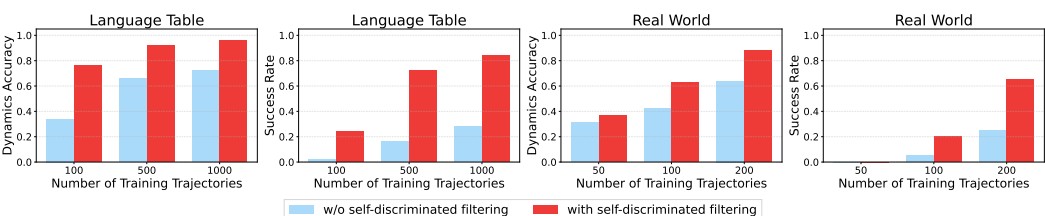

Figure 7: Ablations on self-discriminated filtering.

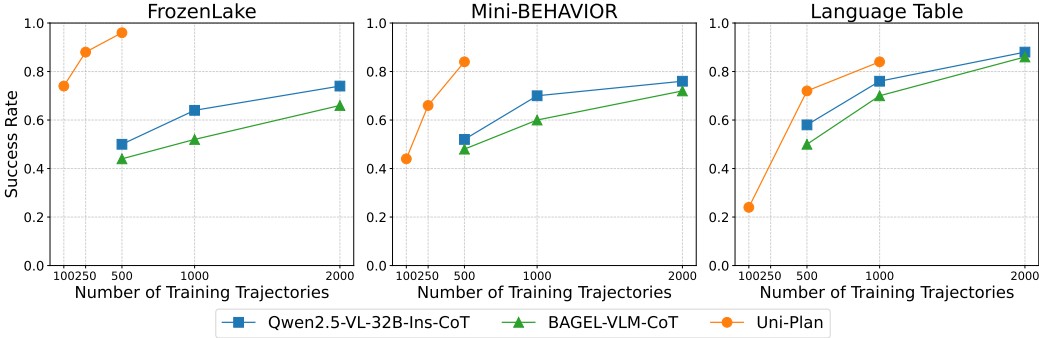

Figure 8: Data scaling trends of two VLM methods: Qwen2.5-VL-32B-Ins-CoT and BAGEL-VLM-CoT, and our method Uni-Plan.

tially higher prediction accuracy than the model trained from scratch. These results indicate that BAGEL's strong generalization stems from its pretrained model and that, with few-shot finetuning, it can serve as a reliable dynamics model for downstream tasks. Additional qualitative comparisons between the two models are provided in Appendix C.1.

The finetuned dynamics model is further enhanced by our proposed *self-discriminated filtering*. Before evaluating its impact, we first verify that this technique is capable of reliably distinguishing correct dynamics predictions from incorrect ones. As shown by confusion matrices in Figure 6, it achieves high accuracy and recall with a low false-positive rate, indicating its strong ability to judge the correctness of its own predicted transitions. Subsequently, we conduct an ablation study to investigate its influence on planning. Figure 7 reports the accuracy of dynamics predictions and the planning success rates with and without the filtering. As shown in Figure 7, the self-discriminated filtering effectively reduces prediction errors in dynamics and, as a result, substantially improves planning success rates.

### 3.3 DATA SCALING WITHOUT EXPERT DEMONSTRATIONS

In this section, we examine the data scalability of our planning system. While existing VLM baselines require expert-collected datasets to adapt their policies (Driess et al., 2023; Mu et al., 2023), our framework can be trained effectively on non-expert trajectories, where actions may be suboptimal. Figure 8 compares scaling trends for Qwen2.5-VL-32B-Ins-CoT, BAGEL-VLM-CoT, and Uni-Plan. Despite relying only on non-expert data, Uni-Plan consistently achieves higher performance with the same amount of data. In particular, just 500 trajectories are sufficient for Uni-Plan to reach strong performance, whereas VLMs fail to achieve competitive results even with four times as much data on *FrozenLake* and *Mini-BEHAVIOR*.

## 4 RELATED WORK

Our work intersects with several research areas, including decision-making with language models, thinking with images, self-verification, and unified multimodal models. We provide a comprehensive discussion below.

**Decision-Making with Language Models.** Motivated by the strong reasoning capabilities of large language models (LLMs) and vision–language models (VLMs), many studies have explored their application to decision-making. Huang et al. (2022a) demonstrate that LLMs can serve as zero-shot planners, decomposing high-level tasks into mid-level plans via prompting. Ichter et al. (2022) augment LLMs with affordances to ground them in real-world robotic tasks. Huang et al. (2022b) further incorporate environment feedback to form an inner monologue, enabling richer planning and control. However, all of these approaches operate solely in the text modality and thus lack grounded perception. To overcome this limitation, several works leverage VLMs for decision-making. For instance, Driess et al. (2023) and Mu et al. (2023) combine LLMs with vision encoders to form VLMs that, after finetuning on embodied datasets, can handle a wide range of embodied planning tasks. Hu et al. (2023) show that advanced closed-source VLMs such as GPT-4V can solve many open-world manipulation tasks without finetuning. Beyond manipulation, Zhang et al. (2024a) demonstrate the use of VLMs for vision-and-language navigation. While these methods have unique advantages, they generally reduce decision-making to treating an LLM/VLM as a policy, and therefore lack counterfactual reasoning ability. In contrast, another line of work formulates decision-making as planning with world models. Hao et al. (2023) and Zhao et al. (2023) repurpose LLMs as both world models and reasoning agents, incorporating principled planning algorithms such as Monte Carlo Tree Search for strategic exploration. More recently, Du et al. (2024) introduce video language planning (VLP), a framework for complex visual tasks in which VLMs act as policies and value functions, and text-to-video models serve as dynamics models. VLP is most relevant to our work since both VLP and our method employ beam search for visual task planning. Nevertheless, our approach differs in key aspects: (i) we unify all roles within a single model, whereas VLP requires two separate models, making our method more efficient at inference; (ii) our model can act as a self-discriminator to reduce hallucinations; and (iii) we demonstrate superior data scalability compared with VLM-based planning.

**Thinking with Images.** The driving idea of our work is that incorporating images into the thinking process can enhance reasoning ability. Many related studies share the same insight and demonstrate effectiveness on reasoning tasks such as mathematics or VQA. Hu et al. (2024) propose the visual chain-of-thought, which generates Python code to invoke external tools for sketching. Zhou et al. (2024) similarly leverage tools for image manipulation to create visual rationales in chain-of-thought reasoning. These methods, however, rely on external modules for image generation. A more promising direction is native multimodal reasoning. For example, Li et al. (2025) finetune a unified multimodal model for multimodal visualization-of-thought, enabling the UMM to produce visualizations of their reasoning traces. Chern et al. (2025) propose iterative refinement of image generation through visual reasoning. Despite these advances, existing approaches use UMMs mainly for visualizing reasoning traces or refining generations, rather than for more sophisticated decision-making as in our work.

**Self-Verification.** Recent studies explore enabling large language models (LLMs) to verify their own outputs. Weng et al. (2023) propose a self-verification strategy that allows large language models (LLMs) to reevaluate their own reasoning to improve answer reliability. Miao et al. (2024) introduce a multi-stage approach that breaks the problem down into a series of simpler tasks and perform step-by-step check. Ma et al. (2025) train models via reinforcement learning to strengthen both self-verification and self-correction abilities. However, these approaches focus purely on textual reasoning. In contrast, our proposed *self-discriminated filtering* extends self-verification to multimodal dynamics prediction, where a UMM generates candidate next observations and verifies them via inverse-dynamics inference, filtering invalid transitions.

More broadly, our self-discriminated filtering can be understood as a form of *consistency regularization* between forward and inverse dynamics (Tarvainen & Valpola, 2017). This echoes established ideas in representation learning, where cycle-consistency or bidirectional prediction serves as a regularizer that improves sample efficiency and robustness (Zhu et al., 2017). In our setting, this principle grounds multimodal generation in action–observation consistency, making the dynamics model both more faithful and more useful for planning (Jordan & Rumelhart, 1992).

While Uni-Plan still inherits the horizon-dependence of model-based planning where errors compound over multi-step rollouts (Talvitie, 2014; Venkatraman et al., 2015), beam search combined with self-discriminated filtering provides a partial remedy by pruning implausible futures early. Moreover, the heuristic value function plays a role analogous to an admissible heuristic in A* search (Hart et al., 1968; Russell & Norvig, 1995), by estimating the number of steps to goal and thereby

prioritizing promising beams and reducing wasted computation on implausible branches. Although our heuristic is learned rather than guaranteed admissible, this connection clarifies how Uni-Plan mitigates horizon-related error accumulation not only through more accurate dynamics but also through structured search.

**Unified Multimodal Models.** Since our method is built on a UMM, it is necessary to review related work in this field. One line of research assembles off-the-shelf specialized LLMs and visual generative models by tuning adapters or learnable tokens, such as NExT-GPT (Wu et al., 2024), DreamLLM (Dong et al., 2024), Seed-x (Ge et al., 2024), and BLIP3-o (Chen et al., 2025a). Alternatively, other work integrates multimodal understanding and generation objectives within a single architecture, including Chameleon (Team, 2024), Show-o (Xie et al., 2025a), Transfusion (Zhou et al., 2025), Emu3 (Wang et al., 2024), Janus-Pro (Chen et al., 2025b), and BAGEL (Deng et al., 2025). We adopt BAGEL as the foundation model in our system because it achieves state-of-the-art performance in both multimodal understanding and image generation among these approaches.

**Model-based Reinforcement Learning.** World models are widely regarded as a powerful means to improve decision-making, a view supported by numerous model-based RL approaches such as Dreamer (Hafner et al., 2020; 2021; 2025) and MuZero (Schrittwieser et al., 2020). However, these methods are typically developed for a single, fixed MDP and must learn dynamics from scratch, which limits their generality and scalability. In contrast, our work builds on the insight that images and language naturally align with the state–action formulation, allowing a pretrained UMM to serve as a general multimodal world model. This perspective enables leveraging large-scale image–language datasets for finetuning, offering a more scalable path toward a generalist model-based RL paradigm.

## 5 CONCLUSION AND LIMITATIONS

In this paper, we presented *Uni-Plan*, a planning framework built on Unified Multimodal Models (UMMs) where a single model simultaneously serves as policy, dynamics model, and value function. The central challenge we identified is learning a faithful dynamics model. To address this, we introduced a self-discriminated filtering mechanism that allows the generative model to act as its own discriminator, filtering out invalid dynamics predictions. Experimental results show that Uni-Plan outperforms VLM-based decision-making paradigms on long-horizon planning tasks, owing to its capacity to function as a highly generalizable dynamics model further reinforced by our proposed filtering method. Uni-Plan also exhibits strong data scalability, requiring no expert demonstrations for fine-tuning and outperforming VLMs when trained with the same amount of data.

Unlike prior approaches that use generated images primarily for visualization of reasoning traces, Uni-Plan employs image generation for *counterfactual reasoning*. By simulating multiple possible futures under different actions, the model does not merely illustrate its thought process but actively evaluates alternative trajectories. This constitutes a shift from visual explanation to visual reasoning as computation, where generated images are intermediates in search and decision-making rather than expository artifacts. We view this as a conceptual leap: images here are not outputs to be consumed by humans but internal representations used by the model to reason about the world.

Our approach nonetheless has limitations. Reasoning through image generation incurs higher computational costs during inference (Table 4), though we expect these to diminish with more efficient foundational models. In addition, the current value function is deliberately simple, producing a scalar step-to-goal estimate; more expressive forms of value prediction could benefit complex domains. Addressing these issues points to a practical research agenda: reducing inference cost, improving value estimation, and exploring broader applications of counterfactual visual reasoning in planning.

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

## A  BRIEF INTRODUCTION TO BAGEL

BAGEL (Deng et al., 2025) is an open-source foundational model that natively supports both multi-modal understanding and generation. In this section, we describe its architecture, pretraining data, and the capabilities of the pretrained model.

The backbone of BAGEL is derived from the Qwen2.5 LLM (Yang et al., 2024). For visual understanding, it employs a Vision Transformer (ViT) encoder to convert raw pixels into visual tokens. For visual generation, BAGEL first applies a pretrained VAE to map images from pixel space to a latent space, and then adopts Rectified Flow (Lipman et al., 2023; Esser et al., 2024) in that latent space to generate images. Text generation is performed autoregressively, whereas image generation proceeds in parallel. In addition, BAGEL adopts a Mixture-of-Transformers (MoT) architecture that uses separate QKV projectors and feed-forward networks (FFNs) for understanding and generation while sharing the same attention layers. Each component is initialized from Qwen2.5-7B, resulting in a total of roughly 14B parameters (only 7B parameters active during inference).

BAGEL is pretrained on interleaved multimodal datasets encompassing multimodal conversation, text-to-image generation, and image manipulation, which enables seamless integration of diverse generative tasks. In the early stages of pretraining, it is primarily trained on simple text-to-image (T2I) and image-to-text (I2T) pairs; later stages introduce high-resolution T2I and I2T pairs as well as interleaved multimodal understanding and generation data. Cross-entropy loss is applied to text tokens, while mean-squared error loss is used for image token generation.

Thanks to this comprehensive training corpus, BAGEL exhibits superior visual understanding and image generation capabilities compared with other leading open-source models (Bai et al., 2025; Chen et al., 2024; Ge et al., 2024; Chen et al., 2025b). More importantly, BAGEL is capable of high-fidelity image editing, which is more challenging than T2I generation because it requires precise control over image details according to textual instructions while maintaining overall visual consistency. This image-editing ability underpins its role as a reliable dynamics model in our planning framework.

## B  IMPLEMENTATION DETAILS

### B.1  DATA COLLECTION

**Simulation tasks.**  There are three simulated environments in our experiments: *FrozenLake*, *Mini-BEHAVIOR*, and *Language Table*. We show some illustrations of these tasks in Figure 9. As for these three simulated tasks, we train RL agents to collect expert trajectories and also randomly sample some non-expert trajectories.

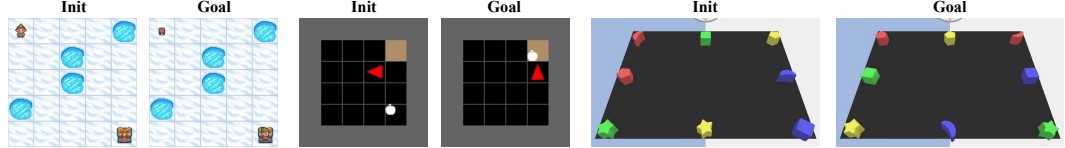

Figure 9: Illustrations of FrozenLake, Mini-BEHAVIOR, and Language Table.

**Real-world tasks.**  For the real-world task (visualized in Figure 3), we collect 200 full-horizon expert demonstrations via human teleoperation. The task requires the robot to execute a long-horizon sequence of three subtasks to rearrange the objects on the table to the goal:

- *Open* or *close the trash can* (if trash-related actions occur).
- *move X on Y to Z* with varying object-container pairs, where *X* denotes the target object and *Y* and *Z* represent the source and the target containers.

As illustrated in Figure 10, we involve unseen objects and containers in the test set for challenging visual discrimination and dynamics prediction. Moreover, when trash manipulation is involved, the

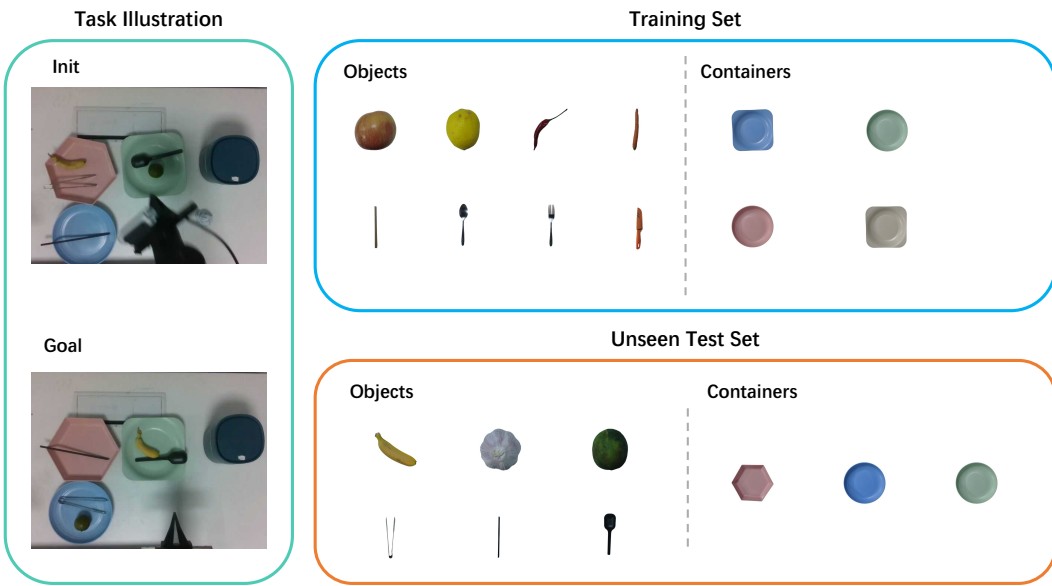

Figure 10: Setting for real-world tasks. The training set contains 8 objects (4 foods and 4 cutlery) and 4 containers, and the test set contains 6 unseen objects and 3 unseen containers.

robot must open the trash can before the first such action and close it after the last, enforcing stateful environmental awareness. Overall, this real-world task demands joint competence in fine-grained object and container recognition as well as accurate dynamics prediction.

**Overall data statistics.** We train Uni-Plan on mixed expert and randomly sampled data while training Qwen2.5-VL (Bai et al., 2025) on pure expert data, and keep the same amount of data during training. Dataset statistics are presented in Table 2.

Table 2: Dataset Statistics for different tasks.

| Tasks | Num of trajectories for training | Num for test | Average Length of trajectories |
|---|---|---|---|
| FrozenLake | 500 | 50 | 7.3 |
| Mini-BEHAVIOR | 500 | 50 | 7.7 |
| Language Table | 500 | 50 | 8.8 |
| Real-world Task | 200 | 20 | 8.2 |

## B.2 IMPLEMENTATION DETAILS OF UNI-PLAN

Uni-Plan finetunes BAGEL on each task using 8×H100 GPUs for 3,000 gradient steps with a constant learning rate of 1e−6, requiring roughly 6 hours of training. During finetuning, the sampling ratio between image-generation data (for the dynamics model) and visual-understanding data (for the policy, value function, and inverse dynamics) is set to 1:1.

As for inference, we list the hyperparameters of Uni-Plan in Table 3. We also present the detailed inference cost in Table 4. Although our model is relatively time-consuming for reasoning, it fortunately requires no replanning and thus needs only a single inference pass.

Table 3: Hyperparameters of beam search.

| Task | Beams | Action Branch | Dynamics Branch |
|---|---|---|---|
| FrozenLake | 2 | 4 | 1 |
| Mini-BEHAVIOR | 2 | 5 | 1 |
| Language Table | 2 | 4 | 4 |
| Real World | 2 | 4 | 8 |

Table 4: Inference cost on one H100 GPU.

| Task | Image Resolution | Images/Step | Time/Step |
|---|---|---|---|
| FrozenLake | 512 × 512 | 8 | 16s |
| Mini-BEHAVIOR | 512 × 512 | 10 | 18s |
| Language Table | 912 × 512 | 32 | 140s |
| Real World | 688 × 512 | 64 | 199s |

To help the model assume different functional roles, we employ tailored prompts. The input prompts used for the various tasks are listed as follows.

---

**Prompt for FrozenLake**

**Dynamics Model**
You are now acting as a **world model** that simulates
environment transitions.
Your task is to predict the **next frame of visual
observation**, given the **current observation image** and the
**current action** taken by the agent.
### Environment description:
You are in a maze environment that contains:
- A character (the agent) that can move
- A gift (the goal)
- Several icy trap tiles, represented in **dark blue**
### Action space:
- "move up"
- "move down"
- "move left"
- "move right"
Each action moves the character exactly one tile in the
indicated direction.
Your task is to **predict the next image** that results from
applying the given action to the current image.
You must:
- Ensure spatial and visual **consistency** of all objects (the
character, gift, traps)

**Policy**
You are a vision-language model with advanced decision-making
abilities.  You will be shown a maze image.
In the maze:
- A character is located at a certain position.
- The goal is to reach the gift location.
- **The path must avoid all hole tiles, which are represented
in dark blue.**
Your task is to carefully observe the image, and then **give
an action for next move** for the character to reach the gift
**without stepping on any hole tiles**.
The available actions are:
- 0:  go left
- 1:  go down
- 2:  go right
- 3:  go up
**Your response must be a json form like:  {"action_id":  0,
"action_name":  "go up"}.  Note that you can only give one of
these four available actions.**

**Value Function**
You are a vision-language model with advanced reasoning
ability.  You will be shown a maze image and determine how many
steps left to reach the goal.
In the maze:
- A character is located at a certain position.
- The goal is to reach the gift location.
- **The path must avoid all hole tiles, which are represented
in dark blue.**
Your task is to carefully observe the image, and then **give
the number of steps** for the current character to reach the

---

```
gift **without stepping on any hole tile**.**Note that if the
character falls into the hole, you should output 100 meaning
that it can never reach the goal**.
In each step, the character can only perform one of the
following actions:
- go left
- go down
- go right
- go up
**Your response must be a json form like: {"num_steps_left":
N}. Do not include any other output.**
```

**Prompt for Mini-BEHAVIOR**

```
Dynamics Model
You are now acting as a **world model** that simulates
environment transitions.
Your task is to predict the **next frame of visual
observation**, given the **current observation image** and the
**current action** taken by the agent.
### Environment description:
You are in a maze environment that contains:
- An agent (the red triangle) that can move
- A table marked by brown region
- An apple
### Action space:
- "turn left"
- "turn right"
- "move forward"
- "pick up apple"
- "drop apple on the table"
### Causal effects of different actions
- turn left: The agent rotates 90 degrees to the left in
place.
- turn right: The agent rotates 90 degrees to the right in
place.
- move forward: The agent moves one step forward in the
direction it is currently facing. Note that the agent cannot
move forward if the cell ahead contains a table or an object.
- pick up apple: The agent picks up the object located in the
cell directly in front of it and carries it. If the object is
in an adjacent cell but the agent is not facing it, the agent
cannot pick it up.
- drop apple on the table: The agent places the object it is
carrying into the table directly in front of it. If the agent
is not facing the table, it cannot place the object.
Your task is to **predict the next image** that results from
applying the given action to the current image.
You must:
- Ensure spatial and visual **consistency** of all objects
- Ensure the causal effect of the given action

Policy
You are a vision-language model with advanced decision-making
abilities.
Your task is to carefully observe the image, and then **give
an action for next move** for the agent to collect the apple to
the table.
### Environment description:
```

```
You are in a maze environment that contains:
- An agent (the red triangle) that can move
- A table marked by brown region
- An apple
### Action space:
- 0:  turn left
- 1:  turn right
- 2:  move forward
- 3:  pick up apple
- 4:  drop apple on the table
### Causal effects of different actions
- turn left:  The agent rotates 90 degrees to the left in
place.
- turn right:  The agent rotates 90 degrees to the right in
place.
- move forward:  The agent moves one step forward in the
direction it is currently facing.  Note that the agent cannot
move forward if the cell ahead contains a table or an object.
- pick up apple:  The agent picks up the object located in the
cell directly in front of it and carries it.
- drop apple on the table:  The agent places the object it is
carrying into the cell directly in front of it.
**Your response must be a json form like:  {"action_id":  0,
"action_name":  "turn left"}.  Note that you can only give one
of these five available actions.**
```

**Value Function**
```
You are a vision-language model with advanced reasoning
abilities.
Your task is to carefully observe the image, and then **give
the number of steps** for the current agent to collect the
apple to the table.  This task needs an agent to first navigate
to the apple to pick it up and then navigate to table to drop
it on.
### Environment description:
You are in a maze environment that contains:
- An agent (the red triangle) that can move
- A table marked by brown region
- An apple
### Action space:
- 0:  turn left
- 1:  turn right
- 2:  move forward
- 3:  pick up apple
- 4:  drop apple on the table
### Causal effects of different actions
- turn left:  The agent rotates 90 degrees to the left in
place.
- turn right:  The agent rotates 90 degrees to the right in
place.
- move forward:  The agent moves one step forward in the
direction it is currently facing.  Note that the agent cannot
move forward if the cell ahead contains a table or an object.
- pick up apple:  The agent picks up the object located in the
cell directly in front of it and carries it.
- drop apple on the table:  The agent places the object it is
carrying into the cell directly in front of it.
```

```
**You should first give the number of steps to pick up the
apple, then the number of steps to drop it on the table,
finally give the total number of steps.  Your response
must be a json form like: {"num_steps_for_pickup": N1,
"num_steps_for_drop": N2, "total_num_steps": N1+N2}.  Do not
include any other output.**
```

**Prompt for Language Table**

**Dynamics Model**
```
You are now acting as a **world model** that simulates
environment transitions.
Your task is to predict the **next frame of visual
observation**, given the following inputs:
- A **current observation image** that shows the current state
of the environment, which may have partial occlusions due to
the robot arm.
- A **natural language instruction** that describes the
intended action.
### Environment description:
You are in a tabletop environment containing exactly **8 unique
objects**, scattered across the table surface.  These objects
differ in both **color** and **shape**:
- blue moon, blue cube
- green star, green cube
- yellow star, yellow pentagon
- red moon, red pentagon
### Important considerations:
- The **instruction** describes the action to be executed
at the current step (e.g., move the blue cube to the red
pentagon).
Your task is to **predict the next image** that results from
applying the given instruction to the current image.
You must:
- Maintain **visual coherence** of the scene (consistent
lighting, robot pose, object appearance)
- Produce a prediction that visually aligns with the expected
effect of the instruction
- Strictly maintain **object consistency**:  the number
of objects must remain exactly the same as in the initial
observation (no missing or extra objects).
```

**Policy**
```
You are a vision-language model with advanced decision-making
abilities.
Your task is to carefully observe the image, and then **give an
action for the next step** to reach the desired goal.
You are given the following inputs:
- A **current observation image** that shows the current state
of the environment.
- A **natural language instruction** that describes the
intended goal.
### Environment description:
You are in a tabletop environment that contains:
- A table surface with **8 unique objects** scattered on it.
These objects vary in both **color** and **shape** and are
categorized as follows:
- blue moon, blue cube
- green star, green cube
```

```
- yellow star, yellow pentagon
- red moon, red pentagon
### Action space:
- move block to block:  move a block to another block (the
target block must be placed at a position with no other objects
around it).
- move block to position:  move a block to a specific position
(where position is one of the 8 positions above, and this
position must be empty.  **If it is occupied, you need to move
the object currently on that position elsewhere first**).
### Causal effects of different actions
- move block to block:  The agent moves a block to another
block.
- move block to position:  The agent moves a block to a
specific position.
**Your response must be a json form like:  {"action":  "xxx"}.
Do not include any other output.*
```

**Value Function**
```
You are a vision-language model with advanced decision-making
abilities.
Your task is to carefully observe the image, and then **give
the number of steps** to reach the desired goal.
You are given the following inputs:
- A **current observation image** that shows the current state
of the environment.
- A **natural language instruction** that describes the
intended goal.
### Environment description:
You are in a tabletop environment that contains:
- A table surface with **8 unique objects** scattered on it.
These objects vary in both **color** and **shape** and are
categorized as follows:
- blue moon, blue cube
- green star, green cube
- yellow star, yellow pentagon
- red moon, red pentagon
In each step, only one object can be moved.  And note that
**the number of left steps will never over 10**.
**Your response must be a json form like:  {"num_steps_left":
N}.  (N<=10) Do not include any other output.**
```

**Inverse Dynamics**
```
You are now acting as an inverse dynamics to infer the action
between two images.
You will be given the following inputs:
- A **current observation image**
- A **next observation image**
### Environment description:
You are in a tabletop environment that contains:
- A table surface with **8 unique objects** scattered on it.
These objects vary in both **color** and **shape** and are
categorized as follows:
- blue moon, blue cube
- green star, green cube
- yellow star, yellow pentagon
- red moon, red pentagon
You need to describe what happened between the two images.
Your output should be like:
```

```
**move {object A} to {object B}.  Do not include any other
output.**
```

## B.3 IMPLEMENTATION DETAILS OF VLMs

**Finetuning Qwen2.5-VL.** We finetune both Qwen2.5-VL-7B and Qwen2.5-VL-32B on each task using $8 \times$ H100 GPUs with a learning rate of $1e-6$. Because the dataset is relatively small (about 500 trajectories containing roughly 4k–5k samples), we iterate over the dataset multiple times until training converges.

We also provide specific prompts to guide these VLMs toward better decision-making. The versions of the prompts without chain-of-thought (w/o CoT), with chain-of-thought (with CoT), and the CoT answer templates used for finetuning are shown below.

---

**Prompt for FrozenLake**

**w/o CoT version**
```
You are a vision-language model with advanced decision-making
abilities.  You will be shown a maze image.
In the maze:
- A character is located at a certain position.
- The goal is to reach the gift location.
- **The path must avoid all hole tiles, which are represented
in dark blue.**
Your task is to carefully observe the image, and then **give
a list of actions ** for the character to reach the gift
**without stepping on any hole tiles**.
The available actions are:
- 0:  go left
- 1:  go down
- 2:  go right
- 3:  go up
**Your response must be a list form like:  ["go right", "go
down", "go left", ...].  Note that you can only give one of
these four available actions.**
```

**with CoT version**
```
You are a vision-language model with advanced decision-making
abilities.  You will be shown a maze image.
In the maze:
- A character is located at a certain position.
- The goal is to reach the gift location.
- **The path must avoid all hole tiles, which are represented
in dark blue.**
Your task is to carefully observe the image, and then **give
a list of actions ** for the character to reach the gift
**without stepping on any hole tiles**.
The available actions are:
- 0:  go left
- 1:  go down
- 2:  go right
- 3:  go up
You should first analyze the positions of traps, then plan the
path step by step, and finally give a list of actions.  Note
that you can only give one of these four available actions.
```

**CoT answer template**
```
<rationale>The traps positions are:  [(1, 2)].
The character is at position (0, 0) and the next move is go
down.
```

```
1188   The character is at position (1, 0) and the next move is go
1189   down.
1190   The character is at position (2, 0) and the next move is go
1191   right.
1192   The character is at position (2, 1) and the next move is go
1193   right.
1194   The character has reached the goal.</rationale>
1195   <action_plan>['go down', 'go down', 'go right', 'go
1196   right']</action_plan>
```

**Prompt for Mini-BEHAVIOR**

**w/o CoT version**
You are a vision-language model with advanced decision-making
abilities.
Your task is to carefully observe the image, and then **give an
action list* for the agent to collect the apple to the table.
### Environment description:
You are in a maze environment that contains:
- An agent (the red triangle) that can move
- A table marked by brown region
- An apple
### Action space:
- 0:  turn left
- 1:  turn right
- 2:  move forward
- 3:  pick up
- 4:  drop
### Causal effects of different actions
- turn left:  The agent rotates 90 degrees to the left in
place.
- turn right:  The agent rotates 90 degrees to the right in
place.
- move forward:  The agent moves one step forward in the
direction it is currently facing.  Note that the agent cannot
move forward if the cell ahead contains a table or an object.
- pick up:  The agent picks up the object located in the cell
directly in front of it and carries it.
- drop:  The agent places the object it is carrying into the
cell directly in front of it.
**Your response must be a list form like:  ["turn right", "move
forward", "pick up", "drop", ...].  Note that you can only give
one of these five available actions.**

**with CoT version**
You are a vision-language model with advanced decision-making
abilities.
Your task is to carefully observe the image, and then **give an
action list* for the agent to collect the apple to the table.
### Environment description:
You are in a maze environment that contains:
- An agent (the red triangle) that can move
- A table marked by brown region
- An apple
### Action space:
- 0:  turn left
- 1:  turn right
- 2:  move forward
- 3:  pick up
```

```
- 4:  drop
### Causal effects of different actions
- turn left:  The agent rotates 90 degrees to the left in
place.
- turn right:  The agent rotates 90 degrees to the right in
place.
- move forward:  The agent moves one step forward in the
direction it is currently facing.  Note that the agent cannot
move forward if the cell ahead contains a table or an object.
- pick up:  The agent picks up the object located in the cell
directly in front of it and carries it.
- drop:  The agent places the object it is carrying into the
cell directly in front of it.
You should first analyze the position of the table, then plan
the path step by step, and finally give a list of actions.
Note that you can only give one of these five available
actions.
```

**CoT answer template**

```
<rationale>The table is at position [0, 3].
The character is at position [1, 2] facing left.  To pick up
the object located at position [3, 3], the next action should
be to turn left.
The character is at position [1, 2] facing down.  To pick up
the object located at position [3, 3], the next action should
be to turn left.
The character is at position [1, 2] facing right.  To pick up
the object located at position [3, 3], the next action should
be to move forward.
The character is at position [1, 3] facing right.  To pick up
the object located at position [3, 3], the next action should
be to turn right.
The character is at position [1, 3] facing down.  To pick up
the object located at position [3, 3], the next action should
be to move forward.
The character is at position [2, 3] facing down.  To pick up
the object located at position [3, 3], the next action should
be pickup.
The character is at position [2, 3] facing down, carrying
the object.  To drop the object on the table, the next action
should be to turn left.
The character is at position [2, 3] facing right, carrying
the object.  To drop the object on the table, the next action
should be to turn left.
The character is at position [2, 3] facing up, carrying the
object.  To drop the object on the table, the next action
should be to move forward.
The character is at position [1, 3] facing up, carrying the
object.  To drop the object on the table, the next action
should be to drop.
The object has been placed on the table.</rationale>
<action_plan>['turn left', 'turn left', 'move forward', 'turn
right', 'move forward', 'pickup', 'turn left', 'turn left',
'move forward', 'drop']</action_plan>
```

**Prompt for Language Table**

**w/o CoT version**
You are a vision-language model with advanced decision-making abilities.
Your task is to carefully observe the current and target goal image, and then **give an action list* for the agent to move the blocks to the desired goal.
### Environment description:
You are in a language table environment that contains:
- 8 blocks:  red moon, red pentagon, blue moon, blue cube, green cube, green star, yellow star, and yellow pentagon.
- 8 absolute positions:  top_center, top_left, top_right, center_left, center_right, bottom_center, bottom_left, and bottom_right.
### Action space:
- move block to block:  move a block to another block (the target block must be placed at a position with no other objects around it).
- move block to position:  move a block to a specific position (where position is one of the 8 positions above, and this position must be empty.  **If it is occupied, you need to move the object currently on that position elsewhere first**).
### Causal effects of different actions
- move block to block:  The agent moves a block to another block.
- move block to position:  The agent moves a block to a specific position.
You should directly give a list of actions.  Note that you can only give one of these two available actions.  Do not give any other text.

**with CoT version**
You are a vision-language model with advanced decision-making abilities.
Your task is to carefully observe the current and target goal image, and then **give an action list* for the agent to move the blocks to the desired goal.
### Environment description:
You are in a language table environment that contains:
- 8 blocks:  red moon, red pentagon, blue moon, blue cube, green cube, green star, yellow star, and yellow pentagon.
- 8 absolute positions:  top_center, top_left, top_right, center_left, center_right, bottom_center, bottom_left, and bottom_right.
### Action space:
- move block to block:  move a block to another block (the target block must be placed at a position with no other objects around it).
- move block to position:  move a block to a specific position (where position is one of the 8 positions above, and this position must be empty.  **If it is occupied, you need to move the object currently on that position elsewhere first**).
### Causal effects of different actions
- move block to block:  The agent moves a block to another block.
- move block to position:  The agent moves a block to a specific position.

```
You should first analyze the target block config, then plan the
path step by step, and finally give a list of actions.  Note
that you can only give one of these two available actions.

CoT answer template
<rationale>Target block config:  'top_center':  ['yellow_star'],
'top_left':  ['red_moon'], 'top_right':  ['green_star'],
'center_left':  ['green_cube'], 'center_right':
['blue_cube'], 'bottom_center':  ['blue_moon'], 'bottom_left':
['yellow_pentagon'], 'bottom_right':  ['red_pentagon'].
Current block config:  'top_center':  ['yellow_star',
'blue_cube'], 'top_left':  ['red_moon'], 'top_right':
['green_star'], 'center_left':  ['green_cube'], 'center_right':
['blue_moon'], 'bottom_center':  [], 'bottom_left':
['yellow_pentagon'], 'bottom_right':  ['red_pentagon'].  Next
action:  move blue_moon to bottom_center.
Current block config:  'top_center':  ['yellow_star',
'blue_cube'], 'top_left':  ['red_moon'], 'top_right':
['green_star'], 'center_left':  ['green_cube'], 'center_right':
[], 'bottom_center':  ['blue_moon'], 'bottom_left':
['yellow_pentagon'], 'bottom_right':  ['red_pentagon'].  Next
action:  move blue_cube to center_right.
Current block config:  'top_center':  ['yellow_star'],
'top_left':  ['red_moon'], 'top_right':  ['green_star'],
'center_left':  ['green_cube'], 'center_right':
['blue_cube'], 'bottom_center':  ['blue_moon'], 'bottom_left':
['yellow_pentagon'], 'bottom_right':  ['red_pentagon'].  Reach
target block config.</rationale>
<action_plan>['move blue_moon to bottom_center', 'move blue_cube
to center_right']</action_plan>
```

**Few-shot Prompting for Closed-source VLMs.** Because the closed-source models cannot be finetuned directly, we employ 10-shot prompting for in-context learning instead. The prompts for these models are identical to those for Qwen2.5-VL, with the addition of a few demonstration examples appended to the end of each prompt.

# C  QUALITATIVE ANALYSIS

## C.1  QUALITATIVE ANALYSIS OF UNI-PLAN

We first present representative successful planning cases of Uni-Plan for each task in Figure 11.

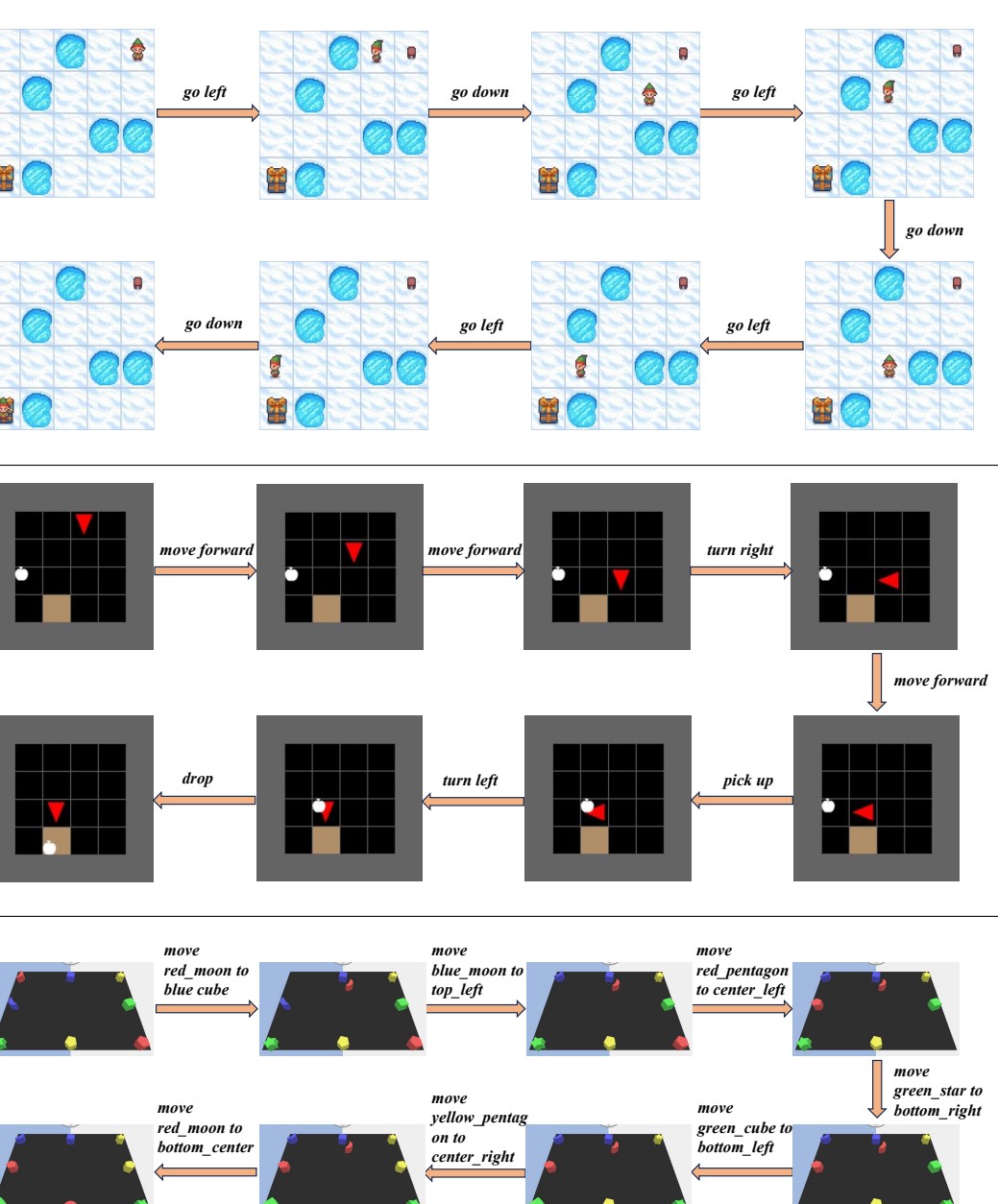

Figure 11: Three planning visualizations with Uni-Plan on FrozenLake, Mini-BEHAVIOR, and Language Table.

Then, we show additional illustrations of dynamics predictions on unseen samples by finetuned BAGEL and BAGEL trained from scratch in Figure 12.

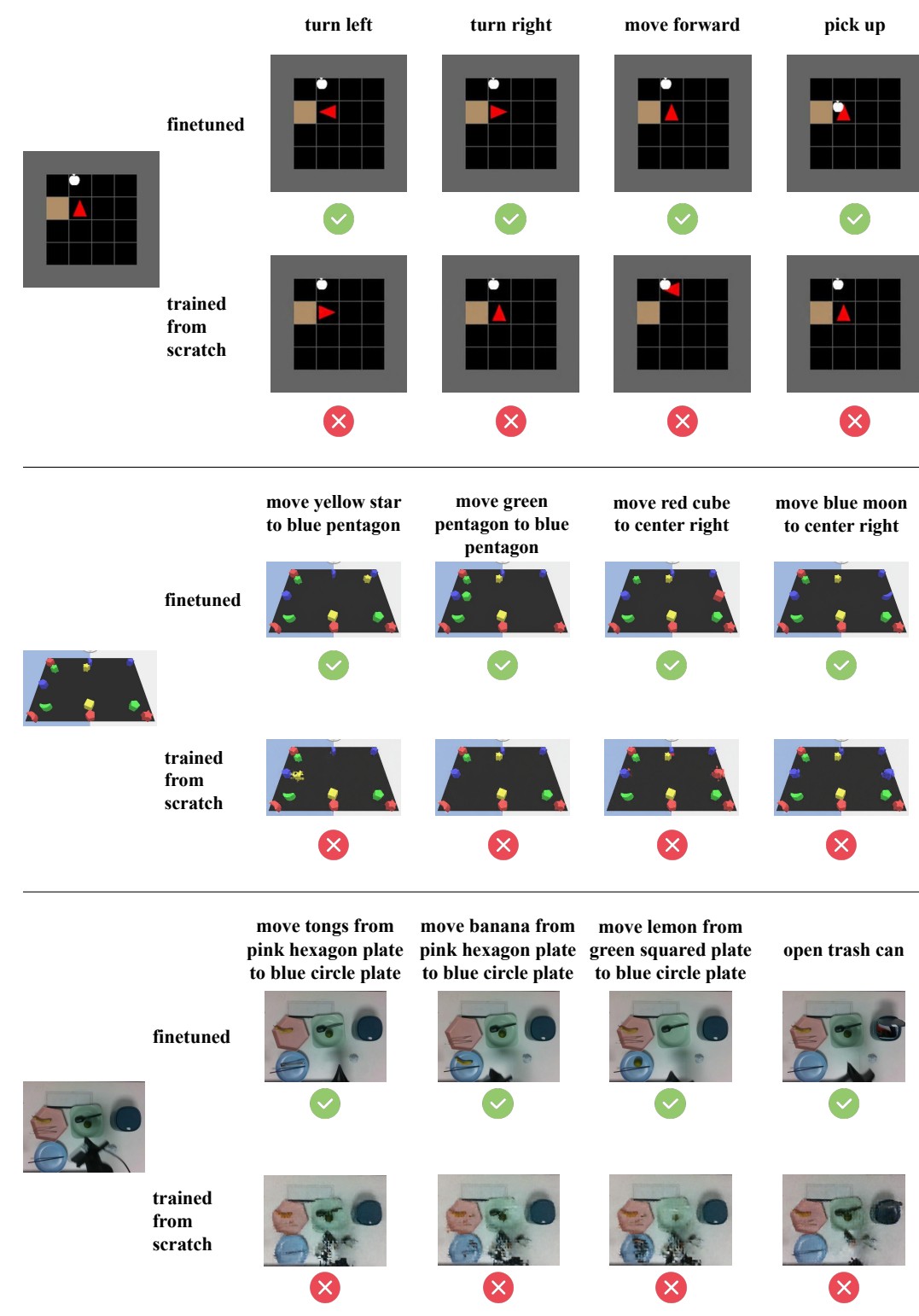

Figure 12: Illustrations of dynamics predictions on unseen samples by finetuned BAGEL and BAGEL trained from scratch

Next, we examine several failure cases, which can be broadly grouped into two categories: *dynamics model errors* and *value function errors*.

**Dynamics Model Errors.**  Figure 13 shows failures caused by incorrect dynamics predictions. The top panel illustrates a wrong placement in the transition, which leads to the next action still trying to move other blocks to *red moon*. The bottom panel shows a case where an object is missing from the predicted observation, causing the policy to continue moving other objects toward that location. Although our proposed *self-discriminated filtering* alleviates such issues, these errors can still occur because only a limited number of predictions are sampled for each state–action pair, and we cannot guarantee that at least one valid prediction will be included, especially in low-data regimes.

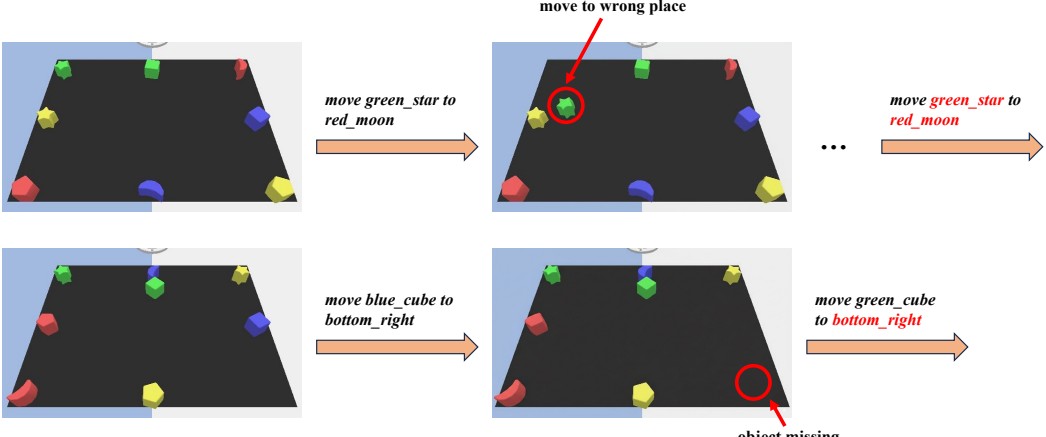

Figure 13: Illustrations of dynamics model errors.

**Value Function Errors.**  Figure 14 depicts failures arising from inaccurate value estimates, which frequently occur in data-scarce regimes, such as when only 100 trajectories are used for finetuning. In the *FrozenLake* task, the value function assigns the best value to the action *move right*. Although the right cell appears closer to the goal, it is actually surrounded by several traps, leaving no path to the goal. In the *Mini-BEHAVIOR* task, the value function favors *turn right* since it wrongly thinks it can go left straight to pick up the object. However that way is blocked by the table. In the *Language Table* task, the value function thinks the task is finished, but the *green star* is not placed in the right position (*top center*) yet.

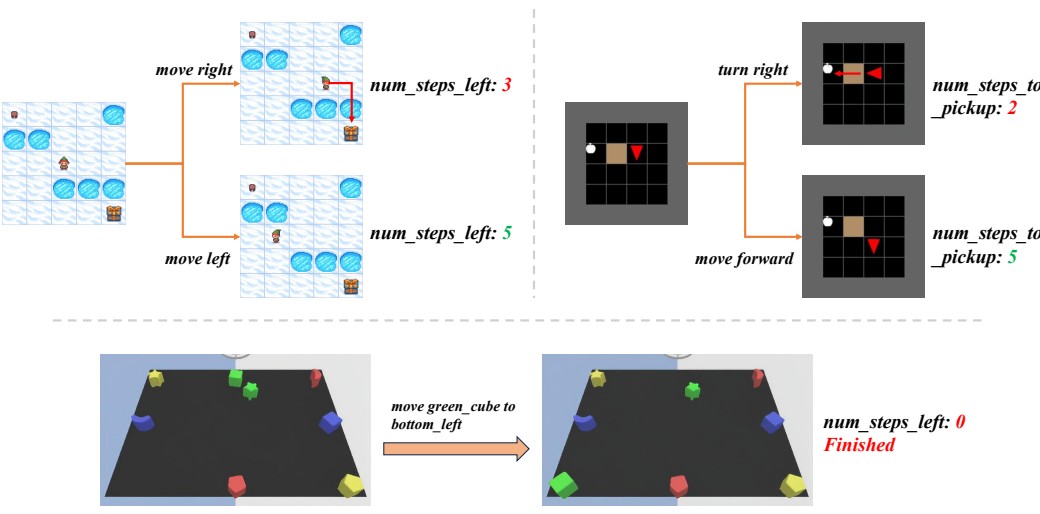

Figure 14: Illustrations of value function errors.

**Violate constraint**

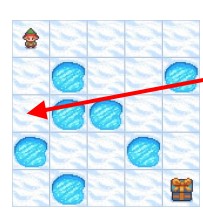

<rationale>The traps positions are: [(1, 1), (1, 4), (2, 1), (2, 2), (3, 0), (3, 3), (4, 1)].
The character is at position (0, 0) and the next move is go down.
The character is at position (1, 0) and the next move is go down.
The character is at position (2, 0) and the next move is go right.
The character is at position (2, 1) and the next move is go right.
The character is at position (2, 2) and the next move is go right.
The character is at position (2, 3) and the next move is go down.
The character is at position (3, 3) and the next move is go right.
The character is at position (3, 4) and the next move is go down.
The character has reached the goal.</rationale>
<action_plan>['go down', 'go down', 'go right', 'go right', 'go right', 'go down', 'go right', 'go down']</action_plan>

**Violate dynamics rule**

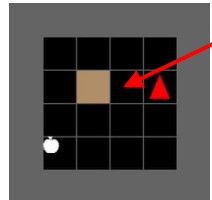

<rationale>The table is at position [1, 1].
The character is at position [1, 3] facing up. To pick up the object located at position [3, 0], the next action should be turn left.
The character is at position [1, 3] facing left. To pick up the object located at position [3, 0], the next action should be move forward.
The character is at position [1, 2] facing left. To pick up the object located at position [3, 0], the next action should be move forward.
The character is at position [1, 1] facing left. To pick up the object located at position [3, 0], the next action should be move forward.
The character is at position [1, 0] facing left. To pick up the object located at position [3, 0], the next action should be turn left.
The character is at position [1, 0] facing down. To pick up the object located at position [3, 0], the next action should be move forward.
The character is at position [2, 0] facing down. To pick up the object located at position [3, 0], the next action should be pickup.
The character is at position [2, 0] facing down, carrying the object. To drop the object on the table, the next action should be turn left.
The character is at position [2, 0] facing right, carrying the object. To drop the object on the table, the next action should be move forward.
The character is at position [2, 1] facing right, carrying the object. To drop the object on the table, the next action should be turn left.
The character is at position [2, 1] facing up, carrying the object. To drop the object on the table, the next action should be drop.
The object has been placed on the table.</rationale>
<action_plan>['turn left', 'move forward', 'move forward', 'move forward', 'turn left', 'move forward', 'pickup', 'turn left', 'move forward', 'turn left', 'drop']</action_plan>

**Violate task rule**

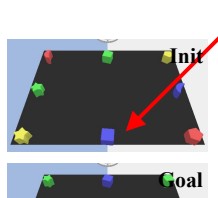

<rationale>Target block config: {'top_center': ['blue_cube'], 'top_left': ['green_star'], 'top_right': ['green_cube'], 'center_left': ['yellow_star'], 'center_right': ['red_pentagon'], 'bottom_center': ['red_moon'], 'bottom_left': ['blue_moon'], 'bottom_right': ['yellow_pentagon']}.
Current block config: {'top_center': ['green_cube'], 'top_left': ['red_moon'], 'top_right': ['yellow_pentagon'], 'center_left': ['green_star'], 'center_right': ['blue_moon'], 'bottom_center': ['blue_cube'], 'bottom_left': ['yellow_star'], 'bottom_right': ['red_pentagon']}.Next action: move blue_cube to red_moon.
Current block config: {'top_center': ['green_cube'], 'top_left': ['red_moon', 'blue_cube'], 'top_right': ['yellow_pentagon'], 'center_left': ['green_star'], 'center_right': ['blue_moon'], 'bottom_center': [], 'bottom_left': ['yellow_star'], 'bottom_right': ['red_pentagon']}.Next action: move red_moon to bottom_center.
Current block config: {'top_center': ['green_cube'], 'top_left': ['red_moon', 'blue_cube'], 'top_right': ['yellow_pentagon'], 'center_left': ['green_star'], 'center_right': ['blue_moon'], 'bottom_center': ['red_moon'], 'bottom_left': ['yellow_star'], 'bottom_right': ['red_pentagon']}.Next action: move blue_cube to bottom_center.
Current block config: {'top_center': ['green_cube'], 'top_left': ['red_moon'], 'top_right': ['yellow_pentagon'], 'center_left': ['green_star'], 'center_right': ['blue_moon'], 'bottom_center': ['red_moon', 'blue_cube'], 'bottom_left': ['yellow_star'], 'bottom_right': ['red_pentagon']}.Next action: move blue_moon to top_center.
Current block config: {'top_center': ['blue_cube', 'blue_moon'], 'top_left': ['red_moon'], 'top_right': ['yellow_pentagon'], 'center_left': ['green_star'], 'center_right': [], 'bottom_center': ['red_moon', 'blue_cube'], 'bottom_left': ['yellow_star'], 'bottom_right': ['red_pentagon']}.Next action: move red_pentagon to center_right.
Current block config: {'top_center': ['blue_cube', 'blue_moon'], 'top_left': ['red_moon'], 'top_right': ['yellow_pentagon'], 'center_left': ['green_star'], 'center_right': ['red_pentagon'], 'bottom_center': ['red_moon', 'blue_cube'], 'bottom_left': ['yellow_star'], 'bottom_right': []}.Next action: move yellow_pentagon to bottom_right.
Current block config: {'top_center': ['blue_cube', 'blue_moon'], 'top_left': ['red_moon'], 'top_right': [], 'center_left': ['green_star'], 'center_right': ['red_pentagon'], 'bottom_center': ['red_moon', 'blue_cube'], 'bottom_left': ['yellow_star'], 'bottom_right': ['yellow_pentagon']}.Next action: move green_cube to top_right.
Current block config: {'top_center': ['blue_cube', 'blue_moon'], 'top_left': ['red_moon'], 'top_right': ['green_cube'], 'center_left': ['green_star'], 'center_right': ['red_pentagon'], 'bottom_center': ['red_moon', 'blue_cube'], 'bottom_left': ['yellow_star'], 'bottom_right': ['yellow_pentagon']}.Reach target block config.</rationale>
<action_plan>['move blue_cube to red_moon', 'move red_moon to bottom_center', 'move blue_cube to bottom_center', 'move blue_moon to top_center', 'move red_pentagon to center_right', 'move yellow_pentagon to bottom_right', 'move green_cube to top_right']</action_plan>

Figure 15: Qualitative analysis of errors of CoT-based planning with Qwen2.5-VL-32B-Ins-CoT on FrozenLake, Mini-BEHAVIOR, and Language Table.

## C.2 QUALITATIVE ANALYSIS OF VLMS

To highlight the limitations of using VLMs for planning, we present several failure cases of Qwen2.5-VL-32B-Ins-CoT in Figure 15.

We identify three distinct factors contributing to these failures. In the *FrozenLake* task, the model violates the safety constraint: it recognizes a trap at position (2,1) but still chooses to move right from (2,0). In the *Mini-BEHAVIOR* task, the plan disregards environment dynamics, attempting to move forward when the agent is at (1,2) despite a table blocking the path. In the *Language Table* task, the plan breaks the task rule that only one object may occupy a given position. By the third

step, the bottom-center cell is already occupied by another block, yet the model attempts to move the blue cube there. We argue that these errors stem from the fact that chain-of-thought reasoning does not explicitly construct a world model to predict the outcomes of different actions or use a reward mechanism to evaluate those outcomes. a limitation also noted in prior work (Hao et al., 2023; Zhao et al., 2023).

## D OMITTED EXPERIMENTS

### D.1 PLANNING UNDER PARTIAL OBSERVATION SCENARIOS

All environments in our main experiment are fully observed without any occlusion in the planning process. To demonstrate that our approach is also able to tackle those partial observation scenarios, we further collect a new real-world dataset in which the robot arm frequently occludes objects during manipulation. Predicting future states under occlusion requires the model to infer the identity and placement of partially or fully hidden objects.

To handle this, we provide the dynamics model with an initial unoccluded observation during prediction, enabling it to reason about occluded objects even when they disappear in subsequent frames. As shown in Figure 16, the model successfully infers and predicts the motion of occluded objects, demonstrating that our method extends to substantially more challenging real-world scenarios.

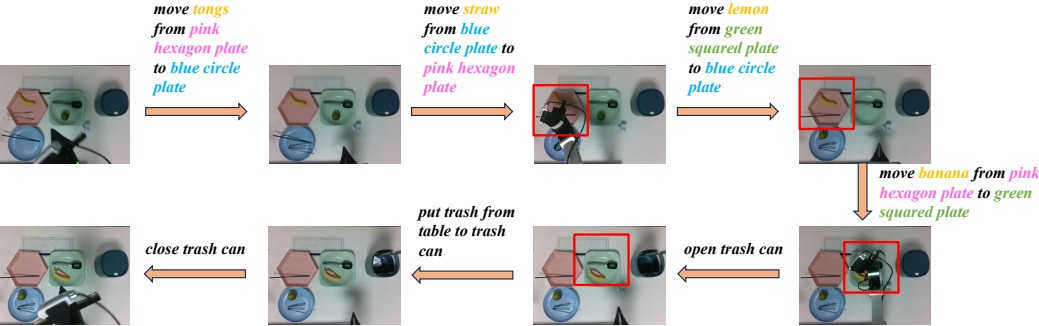

Figure 16: An illustration of planning under the robot's arm occlusion.

### D.2 COMPARISON TO PLANNING APPROACHES WITH LLMS

Besides comparing different VLMs, we also include a classic LLM-based planning baseline, Say-Can (Ichter et al., 2022). SayCan requires iterating over the entire action (skill) space, making it infeasible to evaluate on the Language Table task. Therefore, we omit this setting. The core component of SayCan is an affordance function that determines which actions are feasible in the current state. For FrozenLake, all actions (left / down / right / up) are always valid and thus require no affordance filtering. For Mini-BEHAVIOR, "pick up" is feasible only when the object is not currently held, and "drop" is feasible only when an object is held; we use ground-truth simulator information to implement this affordance logic. For the real-world rearrangement task, we design a small rule-based affordance module to follow the SayCan procedure.

Table 5: Success rates of SayCan and our method.

| Method | Frozen Lake | Mini-BEHAVIOR | Real World |
|---|---|---|---|
| SayCan | 0.32 | 0.22 | 0.40 |
| Uni-Plan | 0.95 | 0.83 | 0.63 |

Table 5 reports the comparison between SayCan and our method. Importantly, SayCan requires online interaction with the environment-executing an action at each step to obtain a new observation. In contrast, our method and the VLM baselines operate in a purely offline manner, generating a

complete plan from a single initial observation. Despite this stricter condition, our method still achieves substantially higher performance.

## E    THE USE OF LARGE LANGUAGE MODELS (LLMs)

Large language models (LLMs) were used in the preparation of this manuscript for sentence-level editing, including improving grammar, clarity, and readability.

