# OpenReview forum: "Planning with Unified Multimodal Models"
_ICLR.cc/2026/Conference — Submitted to ICLR 2026_

### Official Review · Reviewer_rs1q · 2025-10-15

**Soundness:** 3
**Presentation:** 3
**Contribution:** 2
**Rating:** 4
**Confidence:** 5

**Summary:**

The paper argues that unified multimodal models (UMMs) can improve long‑horizon planning by treating a single generative model as policy, dynamics, and value, and by “reasoning through generated visual content.” It further proposes self‑discriminated filtering, where the same generative model filters its own dynamics predictions to reduce hallucinations.

**Strengths:**

- Clear articulation of the limitation of language‑only reasoning for decision‑making and a plausible motivation for multimodal reasoning.
 - The idea of integrating policy, dynamics, and value into a single UMM is conceptually coherent and could simplify interface complexity.
 - Self‑discriminated filtering targets a genuine pain point (hallucinated dynamics) and aims to avoid reliance on external validators.

**Weaknesses:**

- Self‑discriminated filtering may be circular: Having the generator judge its own rollouts can lead to correlated errors. Without an independent criterion (calibration, consistency checks across models, or grounded constraints), filtering may either accept coherent hallucinations or over‑reject plausible futures.
- How is uncertainty addressed in partially observable environments?
- Relation to prior model‑based RL and multimodal planning: The positioning relative to existing unified architectures (e.g., world models that produce latent visuals, diffusion planners with internal consistency checks) is underdeveloped.

**Questions:**

- How does the framework differ from world models that generate latent visuals and from diffusion‑based planners with internal consistency checks?
- How does a UMM that reasons via generated visuals compare to a MuZero‑style latent‑space world model?

---

> ### Author Response · Authors · 2025-11-22
> **Response to Reviewer rs1q (Part 1)**
>
> TL;DR: Based on your valuable suggestions, we
> 1. did quantitative analysis of self-discriminated filtering to demonstrate its reliability as a filtering mechanism (see Response 1);
> 2. added a discussion of prior MBRL approaches in related work (see A1);
> 3. added comparison with MuZero (See A2);
> 4. added experiments of occlusion cases on real-world tasks (see Response 2);
>
> ---
>
> > **Comment 1**: Self‑discriminated filtering may be circular.
>
> **Response 1**: We appreciate the reviewer for raising this question. We first clarify why it is reasonable to use a generative model as a self-discriminator for evaluating its own outputs. Although generation and evaluation rely on the same model, their capabilities are not symmetric. **Large models are typically much stronger at evaluation than at generation**, a phenomenon widely observed in prior self-verification work [1,2,3]. Consequently, **leveraging the model’s superior evaluation ability as a self-discriminator can effectively enhance the quality of its generation.**
>
> To further support this point, we provide quantitative evidence showing that our proposed self-discriminator functions as an effective “filter.” Specifically, we collect 50 positive and 50 negative dynamics predictions from BAGEL, where “positive’’ denotes correct predictions and “negative’’ denotes incorrect ones. We then apply our self-discriminator to classify these samples and compute the resulting confusion matrix.
>
> **Confusion matrix on Language Table task**
> |                | Actual Pos | Actual Neg |
> | -------------- | ---------- | ---------- |
> | **Pred Pos**   | 44         | 2          |
> | **Pred Neg**   | 6          | 48         |
>
> Accuracy: 92%; Recall: 88%; False Positive Rate: 4%
>
> **Confusion matrix on Real World task**
> |                | Actual Pos | Actual Neg |
> | -------------- | ---------- | ---------- |
> | **Pred Pos**   | 49         | 4          |
> | **Pred Neg**   | 1          | 46         |
>
> Accuracy: 95%; Recall: 98%; False Positive Rate: 8%
>
> The results indicate that our method achieves high accuracy and recall, while maintaining a low false-positive rate. This suggests that the self-discriminator is unlikely to discard correct dynamics predictions or retain incorrect ones, demonstrating its reliability as a filtering mechanism.
>
> ---
>
> > **Comment 2**: How is uncertainty addressed in partially observable environments?
>
> **Response 2**: Thank you for the question. In partially observable settings, uncertainty mainly arises when objects become occluded. To handle this, our dynamics model receives the initial unoccluded observation as an additional input during prediction. This allows the UMM to retain information about objects that later become hidden and to correctly infer their identity and location. We validate this in a new real-robot dataset containing frequent arm–object occlusions, and the planning rollouts in **Appendix D.1 (Figure 16, Page 31)** show that the model successfully reasons about occluded objects.
>
> Beyond occlusions during execution, another form of partial observability occurs when the initial observation itself is incomplete, as in many navigation tasks. This setting lies outside the scope of our formulation, which assumes producing a full plan from a single initial observation. In such cases, prior work typically relies on replanning (model-predictive control): planning a short horizon, executing the first few actions, receiving a new observation, and replanning to reduce uncertainty. Our method could similarly be extended with a replanning loop, which we leave for future work.
>
> ---
> [1] Weng et al., "Large language models are better reasoners with self-verification".
>
> [2] Miao et al., "Selfcheck: Using llms to zero-shot check their own
> step-by-step reasoning".
>
> [3] Ma et al., "S^2r: Teaching llms to self-verify and self-correct via reinforcement learning".

---

> ### Author Response · Authors · 2025-11-22
> **Response to Reviewer rs1q (Part 2)**
>
> > **Comment 3**: Relation to prior model‑based RL and multimodal planning is underdeveloped.
>
> **Response 3**: Please see our response to your Q1. And following your suggestion, we have included the discussion of differences between our method and prior MBRL algorithms in the related work in our revised paper.
>
> ---
>
> > **Q1**: How does the framework differ from world models that generate latent visuals and from diffusion‑based planners with internal consistency checks?
>
> **A1**: Our framework differs from both latent-visual world models and diffusion-based planners with internal consistency checks in several key aspects.
>
> * Differences with world models that generate latent visuals.
>     * **Task setting.** Prior latent-space world models such as Dreamer [4] and MuZero [5] are **built for a single fixed MDP with low-dimensional actions.** Our framework performs high-level multimodal planning where actions are natural language, **enabling far broader generality than task-specific MDPs**.
>     * **Training regime.** Existing MBRL approaches are **trained from scratch and require extensive online interaction to learn accurate dynamics.** We instead build on a pretrained UMM, **requiring only 100–500 trajectories of finetuning to obtain a strong dynamics model.**
>     * **Scalability and generality.** Since many tasks align naturally with “**image as observation, language as action**,” our method benefits directly from large-scale multimodal pretraining, providing a scalable foundation for generalist, potentially zero-shot world-model-based planning.
>
> * Differences with diffusion-based planners with internal consistency checks.
>     * Similarly, Diffusion-based planners such as AdaptDiffuser [6] or Diffusion-VF [7] also focus on task-specific MDPs rather than natural-language action spaces, and therefore do not address the broader multimodal planning setting considered in our work.
>     * With respect to internal consistency mechanisms, **these approaches typically rely on additional auxiliary modules**—e.g., inverse/forward dynamics networks or viability filters—to judge or prune candidate trajectories. In contrast, our framework uses the UMM’s own evaluation ability as a self-discriminator, enabling internal consistency checking that is both lightweight and coherent, without introducing extra networks.
>
> ---
>
> > **Q2**: How does a UMM that reasons via generated visuals compare to a MuZero‑style latent‑space world model?
>
> **A2**: MuZero cannot process natural-language actions, and more broadly, there is currently no strong latent-space world model that supports multimodal input (language&image). As a result, **MuZero-style methods must build a new world model from scratch for each downstream task.** In our tasks, FrozenLake and Mini-BEHAVIOR can be abstracted into discrete action spaces, allowing a direct empirical comparison. To ensure fairness, we train an offline variant of MuZero using exactly the same dataset as ours.
>
> The results are shown in the following table. We observe that MuZero performs poorly, and **its success rate decreases as the number of simulations increases.** This indicates that, under limited offline data, **MuZero fails to learn a sufficiently accurate dynamics model to support tree search**, so deeper planning actually amplifies model errors. In contrast, our method starts from a pretrained UMM and requires only light finetuning to obtain a strong dynamics model, achieving much higher success rates. Moreover, **MuZero must replan with environment feedback at each step, whereas our method generates a full plan entirely in imagination.**
>
>
> |                      | Frozen Lake | Mini-BEHAVIOR |
> | -------------------- | ----------- | ------------- |
> | MuZero (num_sim=1)   | 0.42        | 0.38          |
> | MuZero (num_sim=10)  | 0.22        | 0.26          |
> | Uni-Plan             | 0.95        | 0.83          |
>
> ---
> [4] Hafner et al., "Dream to control: Learning behaviors by latent imagination".
>
> [5] Schrittwieser et al., "Mastering atari, go, chess and shogi by planning with a learned model".
>
> [6] Liang et al., "AdaptDiffuser: Diffusion Models as Adaptive Self-evolving Planners".
>
> [7] Ioannidis et al., "Diffusion-based Planning with Learned Viability Filters".

---

> > ### Comment · Reviewer_rs1q · 2025-11-24
> >
> > I appreciate the authors’ detailed responses, which have addressed most of my concerns. However, I have a few remaining points for clarification:
> >
> > - Regarding **A1 (Comparison to Dreamer/MuZero)**: I remain unconvinced that "language as action" presents unique challenges compared to existing high-dimensional action domains. For instance, Dreamer and MuZero have been successfully applied to environments like Minecraft, where the combinatorial action space (keyboard + mouse) is exponential. Could you further elaborate on the specific nuances that distinguish language actions from these high-dimensional settings? I am open to being convinced if there are fundamental differences I am overlooking.
> >
> > - Regarding **Q2 (Reasoning via generated visuals)**: I am curious about the trade-offs between predicting raw images versus latent representations. Predicting latents is often i) computationally more efficient and ii) potentially better suited for handling partial observability or abstracting away irrelevant visual details. While I am not requesting new experiments, I would welcome your insights on why pixel-level prediction is preferable in your specific framework.

---

> > > ### Author Response · Authors · 2025-11-24
> > >
> > > Thank you very much for your timely follow-up. We are glad to hear that our previous responses have addressed most of your concerns.
> > >
> > > **Clarification to A1**: The key insight of our work is not that "language as actions" is more challenging, but that language provides a unified, domain-agnostic action space. This is the main reason why we try to build a image-language based world model (UMM) for planning. Dreamer and MuZero, even though perform very well in complex tasks like Minecraft, still operate within a domain-specific action space. Such action spaces do not naturally extend across domains in the way a language-based action space does.
> > >
> > > In contrast, expressing actions in natural language allows tasks as different as Maze navigation and Language Table manipulation to be represented in the same symbolic action space. This makes it possible to leverage broad multimodal datasets and provides a path toward cross-domain, generalist world models, an ability that is less naturally supported by domain-specific action spaces.
> > >
> > > **Clarification to A2**: We agree that latent prediction is more efficient in principle, but currently there is no strong pretrained latent world model that jointly handles images and language. By predicting raw images, we can directly leverage an open-source pretrained UMM, enabling strong dynamics modeling with very little finetuning data.
> > >
> > > Pixel-level prediction also brings two advantages in our framework:
> > > 	1.	Interpretability — predicted frames are human- and model-interpretable, which is crucial for diagnosing errors.
> > > 	2.	Self-discriminated filtering — our filtering mechanism relies on feeding generated images back into the UMM; latent vectors do not provide a natural way to judge transition correctness.
> > >
> > > Thus, pixel-level prediction sort of sacrifices efficiency, but it allows us to fully exploit the strengths of a pretrained multimodal foundation model.
> > >
> > > We hope this explanation helps clarify the nuances you asked about, and we sincerely appreciate your thoughtful questions. We look forward to your further feedback.

---

> > > > ### Comment · Reviewer_rs1q · 2025-11-25
> > > >
> > > > Thank you for your reply. I will raise my score accordingly.

---

> > > > > ### Author Response · Authors · 2025-11-25
> > > > >
> > > > > We are happy that all your concerns have been addressed. We thank the reviewer for raising the score!

---

### Official Review · Reviewer_1NeT · 2025-10-28

**Soundness:** 2
**Presentation:** 3
**Contribution:** 3
**Rating:** 4
**Confidence:** 3

**Summary:**

This work introduces Uni-Plan, a planning framework using unified multimodal models, as opposed to LLMs or VLMs. They propose one model to act as a policy, dynamics model and value function and introduce a self-discriminated filtering mechanism to filter out hallucinations. They evaluate their planning system against open source and closed source VLM-based planning methods, outperforming open-source benchmark and getting close to the closed source GPT-5-Thinking model, whilst their model has a much smaller number of parameters (14B).

**Strengths:**

1) The authors choose a comprehensive set of baselines for comparison, with both open-source and closed-source models. I particularly appreciate the results showcasing a similar performance to GPT-5-Thinking-Tool, a closed-source model, with a much more significant training dataset and compute budget. This way, authors provide an open alternative to the research community.
2) Compared to related work, such as VLP (video language planning), it proposes a single model for policy, dynamics and value function, thus reducing inference time. As expressed in the conclusion, there is, however, further work to be done to improve inference times, especially if this method aims to be applied to real-world planning scenarios.
3) Proposes a self-discriminator filtering mechanism to address and reduce hallucinations and strengthen the dynamics model quality. I also appreciated the additional object-count consistency check to further validate observation predictions.

**Weaknesses:**

1) Given it's a key component in the self-discriminated filtering mechanism, I would've liked to see a more detail on the inverse dynamics inference method used.
2) The authors state that the system addresses long-horizon planning tasks, but I could not see any definition of the choice of hyperparameter `H` for time-horizon and in all 3 environments it looks like the planning task is in the order of tens of steps or less. It would be good if authors could elaborate more on what they regard as long-horizon.
3) Even though I really appreciate the inclusion of the real-world planning example with the dual-arm robot in Section 3.1, I fear only one example in Fig 3 would not suffice to make a claim the planning framework applies reliably to such scenarios. I would've expected a quantitative analysis on the task (FID between imagined state and real execution).
4) Similarly, I would've liked to see numbers on the benefit of self-discriminating filtering across all environments (not just language task) in Fig 5. I'm not sure if they might be included in the appendix.
5) In Fig 4, where authors showcase finetuned BAGEL is a strong dynamics model, this is illustrated only as 1 short example (4 steps) - I believe a quantitative evaluation across all environments and multiple runs would make the claim stronger.

**Questions:**

-  *Suggestion*: There should be a small correction in section 3.3 - authors state Uni-Plan consistently achieves higher performance with fewer trajectories, but in Fig 6, on the language table, the performance on Uni-Plan with 500 traj is on par with (if not a bit lower than) the performance of Qwen2.5 with 1K trajectories. In the abstract and the conclusion, the claim is corrected to state "outperforming VLMs when trained with the **same** amount of data".
- *Suggestion*: It would be good to add the number of parameters for BAGEL-VLM in `Table 1` for clarity and completeness in the comparison.
- *Question*: Could the authors elaborate more on how the hyperparameters A, D, B and H are chosen? I see in the appendix the beams B are 2 for all environments - what is the rationale for choosing that value?

---

> ### Author Response · Authors · 2025-11-22
> **Response to Reviewer 1NeT (Part 1)**
>
> TL;DR: Based on your valuable suggestions, we mainly added the following experiments:
>
> 1. a quantitative analysis on the real-world tasks (see Response 3);
> 2. extra ablation study of self-discriminated filtering on real-world tasks (see Response 4);
> 3. a quantitative evaluation of 'Strong dynamics model' across all environments; (see Response 5).
>
> ---
>
> > **Comment 1**: See a more detail on the inverse dynamics inference method used.
>
> **Response 1**: Thank you for pointing this out. The inverse dynamics component in our self-discriminated filtering mechanism is implemented in a very direct manner by treating the UMM as a standard VLM. Specifically, we feed the model two consecutive observations $(o_t, o_{t+1})$ and ask it to generate the corresponding natural-language action. During training, we decompose each trajectory into $(o_t, o_{t+1}, a_t)$ tuples and supervise the model to predict $a_t$ given the two frames. No additional architecture or optimization trick is required.
>
> ---
>
> > **Comment 2**: Definition of the choice of hyperparameter H for time-horizon & Elaborate more on what they regard as long-horizon.
>
> **Response 2**: We apologize for the confusion caused by the pseudocode. The hyperparameter H is not a fixed hyperparameter in our framework. During beam search, the model uses the predicted “number of steps left” as a heuristic value, and planning automatically terminates when this value reaches zero. To prevent rare cases where the model repeatedly believes the task is not finished, we assign each environment a maximum allowed planning horizon. We have modified the pseudocode to avoid this confusion.
>
> Regarding what we consider “long-horizon,” we follow the conventions in prior high-level embodied planning work (e.g., SayCan [1], PaLM-E [2], EmbodiedGPT [3]), where long-horizon typically refers to 5–10 high-level steps. This differs from low-level control tasks, where a horizon may span hundreds of motor commands, because each high-level step in our setting corresponds to a semantically meaningful subgoal (“move object A to B", etc.). Under this definition, our tasks fall into the long-horizon category: the longest reasoning chain in our tasks reaches 15 high-level steps, which exceeds the planning length used in most existing high-level embodied planning benchmarks.
>
> ---
>
> > **Comment 3**: A quantitative analysis on the real-world tasks.
>
> **Response 3**: Thank you for the suggestion. We agree that additional quantitative evaluation is important for establishing reliability in real-world scenarios. Instead of FID, we provide more direct task-level measurements.
>
> First, we report the dynamics prediction accuracy on the real-world task. After applying our self-discriminated filtering, the model achieves high accuracy, indicating that the imagined transitions align well with real executions.
>
> |  | Dynamics Accuracy on Real World Task |
> | -------- | -------- |
> | BAGEL trained from scratch    | 0.00     |
> | fintuned BAGEL     | 0.64     |
> | fintuned BAGEL with self-discriminated filtering     | 0.88     |
>
> Second, to directly assess the applicability of our planning framework, we report the planning success rates of different models on the real-world task.
>
> |  | Planning Success Rates |
> | -------- | -------- |
> | GPT-5                  | 0.10      |
> | GPT-5-Thinking         | 0.80      |
> | GPT-5-Thinking-Tool    | 0.80      |
> | Qwen2.5-VL-7B-Ins      | 0.07±0.02 |
> | Qwen2.5-VL7B-Ins-CoT   | 0.22±0.02 |
> | Qwen2.5-VL-32B-Ins     | 0.18±0.06 |
> | Qwen2.5-VL-32B-Ins-CoT | 0.33±0.02 |
> | BAGEL-VLM (7B)         | 0.09±0.04 |
> | BAGEL-VLM (7B)         | 0.26±0.02 |
> | Uni-Plan (14B-A7B)     | 0.63±0.02 |
>
> While GPT5-Thinking (Tool) achieves the highest score, our method still surpasses all other open-source models by a substantial margin. Considering that **we finetune using only 200 trajectories and evaluate on challenging unseen scenarios (plase see Appendix B.1 for more details)**, we view this level of performance as highly encouraging.
>
> ---
> [1] Ahn et al., "Do as I can, not as I say: Grounding language in robotic affordances".
>
> [2] Driess et al., "Palm-e: An embodied multimodal language model".
>
> [3] Mu et al., "Embodiedgpt: Vision-language pre-training via embodied chain of thought".

---

> ### Author Response · Authors · 2025-11-22
> **Response to Reviewer 1NeT (Part 2)**
>
> > **Comment 4**: Benefit of self-discriminating filtering across all environments.
>
> **Response 4**: Following your suggestion, we have added an ablation study of the self-discriminating filtering module on the real-world task, and the updated results are included in the revised manuscript. For your convenience, we also report the results below:
>
> | Dynamics accuracy on Real World task | w/o sdf | with sdf |
> | -------- | -------- | -------- |
> | 50 trajs     | 0.31     | 0.37     |
> | 100 trajs     | 0.42     | 0.63     |
> | 200 trajs     | 0.64     | 0.88     |
>
> | Success rate on Real World task | w/o sdf | with sdf |
> | -------- | -------- | -------- |
> | 50 trajs     | 0.00     | 0.00     |
> | 100 trajs     | 0.05     | 0.20     |
> | 200 trajs     | 0.25     | 0.65     |
>
> We would like to clarify that we do not apply self-discriminating filtering on FrozenLake or Mini-Behavior. As explained in Section 2.2, the UMM (BAGEL) becomes sufficiently accurate on these deterministic-dynamics environments after finetuning, making the additional filtering unnecessary for further improving the dynamics model. For these tasks, the main bottleneck instead lies in value estimation, which suggests a promising future direction for exploring how to better leverage UMMs for decision making.
>
> ---
>
> > **Comment 5**: A quantitative evaluation of 'Strong dynamics model' across all environments.
>
> **Response 5**: To provide a quantitative evaluation of BAGEL trained from scratch and finetued BAGEL as dynamics models, we collect 100 unseen samples and test the dynamics accuracy on each task. The results are shown below.
>
>
> |                      | Frozen Lake | Mini-BEHAVIOR | Language Table | Real World|
> | -------------------- | ----------- | ------------- |----------------|-----------|
> | trained from scratch | 0.23        | 0.40          | 0.00           | 0.00      |
> | finetuned            | 0.99        | 0.93          | 0.72           | 0.64      |
>
> It can be observed that the finetuned model is much better than one trained from scratch especially on challenging tasks like Language Table and Real World. We also included this result in **Figure 5, Page 7** of our revised manuscript.
>
>
> ---
>
> > **Suggestion 1**: correct the claim to state "outperforming VLMs when trained with the same amount of data".
>
> **Response**: Thank you for pointing this out. We have corrected this claim in our revised manuscript.
>
> ---
>
> > **Suggestion 2**: Add the number of parameters for BAGEL-VLM in Table 1.
>
> **Response**: Thank you for this suggestion. We have added this in our revised manuscript.
>
> ---
>
> > **Q1**: Could the authors elaborate more on how the hyperparameters A, D, B and H are chosen?
>
> **A1**: The hyperparameters A, D, B, and H are chosen based on simple task-dependent heuristics:
> * $A$ matches the discrete action space for FrozenLake and Mini-BEHAVIOR; for open-ended language-action tasks (Language Table, Real-World), we use A=4 to balance diversity and efficiency.
> * $D$ reflects dynamics difficulty: D=1 for simple tasks (FrozenLake and Mini-BEHAVIOR), and larger values (4 for Language Table and 8 for Real World) for more complex settings to ensure correct transitions appear in the candidate set.
> * $B$ is fixed to 2 across all tasks, which we find sufficient and efficient.
> * $H$ is a safety cap set to the maximum plausible task length; planning itself terminates automatically when the predicted “steps remaining’’ reaches zero.

---

### Official Review · Reviewer_VZiD · 2025-10-30

**Soundness:** 2
**Presentation:** 3
**Contribution:** 3
**Rating:** 4
**Confidence:** 3

**Summary:**

This paper presents Uni-Plan, a planning framework built on unified multimodal models (UMMs) that jointly handle text and image reasoning for decision-making. Unlike prior language- or vision-language-based planners that rely mainly on textual reasoning, Uni-Plan employs a single UMM as the policy, dynamics model, value function, and self-discriminator. A key innovation is the proposed self-discriminated filtering mechanism, where the model validates its own predicted dynamics by comparing inferred inverse actions with executed ones, effectively reducing hallucinated transitions. Experiments on long-horizon planning benchmarks demonstrate that Uni-Plan achieves about 30% higher success rates than open-source VLM-based methods, matches the performance of the GPT-5-Thinking model, and shows strong data scalability without requiring expert demonstrations.

**Strengths:**

1. The paper convincingly demonstrates that a unified multimodal model can serve simultaneously as (i) a policy, (ii) a dynamics model, (iii) a self-discriminator, and (iv) a value function. This is a very interesting idea, and the paper provides solid evidence of its effectiveness.
2. Under this unified setting, the UMM framework significantly boosts the performance of the base VLM, showing clear advantages in planning and reasoning.

**Weaknesses:**

The experimental settings are relatively simple. All real-world and simulated environments are fully observed and easy to predict. I believe the main challenges in this domain lie in semantic understanding, which is largely powered by the base model. For example, a sufficiently strong base model such as GPT-5-Thinking-tool could outperform even a carefully designed architecture like the one proposed in this paper.

**Questions:**

1. Could you further demonstrate the dynamic model’s ability in more unseen environments, or under more challenging real-world conditions?
2. Could you provide additional real-robot experiments, such as pick-and-place tasks involving occlusion or unseen objects?

---

> ### Author Response · Authors · 2025-11-22
> **Response to Reviewer VZiD**
>
> TL;DR: Based on your valuable suggestions, we mainly added the following experiments:
>
> 1. qualitative and quantitative evaluation on more challenging real-world tasks (see A1);
> 2. evaluation on real-world tasks involving occlusion (see A2).
>
> ---
>
> > **Comment 1**: The experimental settings are relatively simple. All real-world and simulated environments are fully observed and easy to predict.
>
> **Response 1**: We appreciate the reviewer’s comment. We agree that evaluating our method under more challenging conditions, such as unseen environments, occlusions, or settings requiring precise semantic understanding, can further demonstrate the strength and generality of our approach. Following your suggestion, we have added qualitative and quantitative evaluations in significantly more difficult real-world scenarios. The details are addressed in our responses to Q1 and Q2 below.
>
> ---
>
> > **Q1**: Could you further demonstrate the dynamic model’s ability in more unseen environments, or under more challenging real-world conditions?
>
> **A1**: To assess the dynamics model under more difficult conditions, we design a substantially more challenging real-world rearrangement-and-cleanup task. The environment contains **4 containers, 5 objects, and 0–2 pieces of randomly placed trash**, and the **training set consists of only 200 trajectories**. Importantly, we intentionally **introduce unseen objects, unseen containers, and unseen backgrounds in the test set.** This requires the dynamics model to accurately recognize and track objects it has never seen before, while also remaining robust to irrelevant background variations, making it a strong test of generalization. Task details are provided in **Figure 10, Page 16** of the revised manuscript.
>
> We first evaluate the dynamics model on this harder benchmark. Results are shown below.
>
> |  | Dynamics Accuracy on Real World Task |
> | -------- | -------- |
> | BAGEL trained from scratch    | 0.00     |
> | fintuned BAGEL     | 0.64     |
> | fintuned BAGEL with self-discriminated filtering     | 0.88     |
>
> Finetuning alone yields only 64% prediction accuracy, demonstrating the difficulty of this task. However, applying our self-discriminated filtering improves the accuracy to 88%. We further evaluate the success rates across different methods. Example qualitative rollouts from this new scenario are shown in **Figure 3, Page 7** of the revised manuscript.
>
> |  | Planning Success Rates |
> | -------- | -------- |
> | GPT-5                  | 0.10      |
> | GPT-5-Thinking         | 0.80      |
> | GPT-5-Thinking-Tool    | 0.80      |
> | Qwen2.5-VL-7B-Ins      | 0.07±0.02 |
> | Qwen2.5-VL7B-Ins-CoT   | 0.22±0.02 |
> | Qwen2.5-VL-32B-Ins     | 0.18±0.06 |
> | Qwen2.5-VL-32B-Ins-CoT | 0.33±0.02 |
> | BAGEL-VLM (7B)         | 0.09±0.04 |
> | BAGEL-VLM (7B)         | 0.26±0.02 |
> | Uni-Plan (14B-A7B)     | 0.63±0.02 |
>
> ---
>
> > **Q2**: Could you provide additional real-robot experiments, such as pick-and-place tasks involving occlusion or unseen objects?
>
> **A2**: Following your suggestion, we additionally evaluate our approach on real-world tasks involving occlusion, a setting that poses major challenges for dynamics prediction. To this end, **we collect a new dataset in which the robot arm frequently occludes objects during manipulation**. Predicting future states under occlusion requires the model to infer the identity and placement of partially or fully hidden objects.
>
> To handle this, we provide the dynamics model with an initial unoccluded observation during prediction, enabling it to reason about occluded objects even when they disappear in subsequent frames. We visualize representative planning trajectories in **Figure 16, Page 31** of the revised manuscript. As shown, the model successfully infers and predicts the motion of occluded objects, demonstrating that our method extends to substantially more challenging real-world scenarios.

---

### Official Review · Reviewer_Lo2k · 2025-10-31

**Soundness:** 2
**Presentation:** 2
**Contribution:** 2
**Rating:** 2
**Confidence:** 3

**Summary:**

This paper introduces a high-level planning method that uses foundation models, specifically unified multi-modal models (UMMs). The UMM is used in four distinct roles: 1) as a policy, 2) as a dynamics model, 3) self-discriminator, and 4) value function.

The proposed method (Uni-Plan) utilizes an image of the environment's initial state and a textual description of the goal to produce a multi-step plan, where each step is executed by a low level controller (the learning of this controller is outside of the paper's scope).

At each step, the method involves 1) sampling A actions from the policy, 2) generating D next observations, and 3) filtering the D next observations down to 1 observation by tracing which action could have led to the next observations and matching it to the actual action. The authors do this for B beams.

**Strengths:**

1) The novel contributions include: 1) using a UMM, 2) fine-tuning with non-expert trajectories, and 3) using a UMM as a self-discriminator. These are all interesting contributions.
2) The proposed method shows high performance with much fewer trajectories—these results are promising!

**Weaknesses:**

1) The authors should cite the published version of the papers in their references (e.g., "Do as I can, not as I say: Grounding language in robotic affordances" was published at CoRL in 2022; "Inner monologue: Embodied reasoning through planning with language models" was published in CoRL in 2022; etc.).
2) The images of figure 3 are slightly out of focus and there is a black edge at the top of each photo that makes them appear unaligned horizontally.
3) No true empirical comparison to prior planning approaches with LLMs/VLMs. Experiments only vary the model type (VLM vs. UMM) and model size. Authors claim to have prior planning approaches as baselines, but I would argue that their baselines are VLM-based plan generators not prior approaches. For example, a good baseline for comparison would be SayCan [Ahn et al., 2022].
4) No statistical rigour.

**Questions:**

1) How many seeds/trials did you run? I see no mention of this and I see no reported standard error or standard deviation.
2) Feel free to address any weaknesses.

---

> ### Author Response · Authors · 2025-11-22
> **Response to Reviewer Lo2k (Part 1)**
>
> TL;DR: Based on your valuable suggestions, we mainly added the following experiments:
>
> 1. an empirical comparison to SayCan (see Response 3).
> 2. run multiple seeds to make the evaluation more rigorous (see Response 4);
>
> ---
>
> > **Comment 1**: Should cite published version of papers.
>
> **Response 1**: Thank you for pointing this out. We have carefully checked each citation in the reference and updated all entries to their corresponding published versions.
>
> ---
>
> > **Comment 2**: The images of figure 3 are slightly out of focus and have black edges.
>
> **Response 2**: We apologize for this issue, which was likely caused by our oversight during the data collection process. To address this, we have collected new real-world data and retrained our model accordingly. The images in Figure 3 have been replaced with the updated results in the revised manuscript.
>
> ---
>
> > **Comment 3**: No true empirical comparison to prior planning approaches with LLMs/VLMs (e.g. SayCan).
>
> **Response 3**: We appreciate the reviewer’s concern and we think some clarification is needed. **The key reason why earlier works designed specialized planning approaches for LLMs is that LLMs initially lacked any visual perception capability.** As a result, methods such as SayCan [1] had to introduce additional components, like affordance models, to compensate for the missing visual grounding. This reliance on external tools was not a conceptual requirement of high-level planning itself, but rather a workaround for the absence of visual understanding in early LLMs.
>
> Subsequent works (e.g., PaLM-E [2], EmbodiedGPT [3]) addressed this limitation by augmenting LLMs with visual encoders, effectively creating early VLMs capable of performing multimodal high-level planning directly. More recently, Hu et al.[4] showed that a strong VLM such as GPT-4V can plan without any auxiliary modules at all and still surpass prior systems like SayCan. **These developments highlight that a pure VLM—without additional task-specific design—has become a standard and competitive baseline for embodied planning.**
>
> Following your suggestion, we include a comparison with SayCan. Notably, SayCan requires online interaction with the environment (executing an action at each step to retrieve the next observation), whereas both our method and the VLM baselines operate in a purely offline manner, producing complete plans from a single initial observation. Even under this more challenging offline-only setting, our method still achieves substantially stronger performance. **The experimental details and results are provided in Appendix D.2 (Page 31) of the revised manuscript.** Note that SayCan needs to iterate the whole action (skill) space and it is not feasible for Language Table so we exclude the evaluation of SayCan on this task.
>
>
> |                              | Frozen Lake | Mini-Behavior | Real World |
> | ---------------------------- | ----------- | ------------- | ---------- |
> | SayCan (interation-based)    | 0.32        | 0.22          | 0.40       |
> | Uni-Plan (interation-free)   | 0.95        | 0.83          | 0.63       |
>
> ---
> [1] Ahn et al., "Do as I can, not as I say: Grounding language in robotic affordances".
>
> [2] Driess et al., "Palm-e: An embodied multimodal language model".
>
> [3] Mu et al., "Embodiedgpt: Vision-language pre-training via embodied chain of thought".
>
> [4] Hu et al., "Look before you leap: Unveiling the power of GPT-4V in robotic vision-language planning".

---

> ### Author Response · Authors · 2025-11-22
> **Response to Reviewer Lo2k (Part 2)**
>
> > **Comment 4**: No statistical rigour.
>
> **Response 4**: To make our results more rigorous, we retrained each model with 3 random seeds and updated Table 1 in the revised manuscript with the corresponding mean and standard deviation.
>
>
>
> |  | Frozen Lake | Mini-BEHAVIOR | Language Table | Real World |
> | -------- | -------- | -------- | -------- |-------- |
> | Qwen2.5-VL-7B-Ins     | 0.33±0.04 | 0.00±0.00 | 0.14±0.03 | 0.07±0.02 |
> | Qwen2.5-VL-7B-Ins-CoT | 0.37±0.02 | 0.15±0.02 | 0.36±0.01 | 0.22±0.02 |
> | Qwen2.5-VL-32B-Ins    | 0.43±0.05 | 0.07±0.03 | 0.28±0.06 | 0.18±0.06 |
> | Qwen2.5-VL-32B-Ins-CoT| 0.49±0.02 | 0.51±0.01 | 0.57±0.02 | 0.33±0.02 |
> | BAGEL-VLM (7B)        | 0.38±0.05 | 0.01±0.02 | 0.23±0.02 | 0.09±0.04 |
> | BAGEL-VLM-CoT (7B)    | 0.43±0.02 | 0.48±0.00 | 0.51±0.02 | 0.26±0.02 |
> | Uni-Plan (14B-A7B)    | 0.95±0.01 | 0.83±0.01 | 0.73±0.02 | 0.63±0.02 |
>
>
> Overall, the variance is very small (typically 0.0–0.06), which is expected given that all models are trained via supervised learning. Moreover, we observe that both CoT and our method exhibit smaller variance (0.01-0.02) compared to the Non-CoT baselines (0.0-0.06).
>
> We attribute this difference to the fact that Non-CoT models produce the final answer in a single forward pass, making them more sensitive to small perturbations in training. In contrast, CoT-based models and ours perform step-by-step reasoning, which provides a more structured intermediate trajectory and leads to more stable predictions across different seeds.
>
> ---
>
> > **Q1**: How many seeds/trials did you run?
>
> **A1**: As noted in Response 4, **we retrained each model with 3 different random seeds**. For each trained model, **we evaluate its performance across 50 test tasks**. For the real-world setup, we evaluate on **20 distinct tasks** due to the operational cost. We believe this level of evaluation is sufficiently comprehensive to demonstrate the advantages and robustness of our proposed method.

---

### Author Response · Authors · 2025-11-22
**General Response**

We would like to express our sincere gratitude to the reviewers for their valuable comments and suggestions that have greatly helped us improve our work. We are greatly encouraged by the positive comments of reviewers, e.g.,

* The paper proposes an interesting idea with novel contributions. (Lo2k, VZiD)
* The experimental results are promising. (Lo2k)
* The proposed self-discriminator filtering mechanism targets a genuine pain point and is a good way to address and reduce hallucinations. (1NeT, rs1q)

After reading these valuable comments, we think most reviewers' concerns are about the insufficient experiments. Therefore, we took every comment seriously and have added extensive and comprehensive new experiments accordingly:

* We added **both qualitative and quantitative evaluation on more challenging real-world tasks**. (VZiD and 1NeT).
* We included **more comprehensive quantitative analysis of "strong dynamics model" and self-discriminated filtering** to support our claims in Section 3.2. (1NeT)
* We **ran multiple seeds for each method** to make the main results more rigorous. (Lo2k)
* We added an experiment to show that **our proposed self-discriminated filtering is a reliable filtering mechanism**. (rs1q)
* We did **empirical comparisons with SayCan and MuZero**. (Lo2k and rs1q)

---

> ### Author Response · Authors · 2025-11-29
> **Follow-up Summary for the Area Chair**
>
> Given the unexpected interruption of the rebuttal phase, we provide this summary to ensure the AC can clearly see each reviewer's main concerns and how we have addressed them in the revised paper.
> Below, we summarise each reviewer’s main concern along with the corresponding modifications we have already incorporated into the revised manuscript.
>
> ---
>
> ### Reviewer Lo2k
> **Concerns:**
> * No true empirical comparison to prior planning approaches like SayCan.
> * No statistical rigour.
>
> **Our revision:**
> * We added an empirical comparison to SayCan (**Appendix D.2, Page 31**) and also explained why VLMs have become standard and competitive baselines for embodied planning (**see Response 3 to reviewer Lo2k**).
> * We ran multiple seeds to make the evaluation more rigorous (**Table 1, Page 6**).
>
> ---
>
> ### Reviewer VZiD
> **Concerns:**
> * The experimental settings are relatively simple. The reviewer wants to see evaluations on more challenging real-world tasks.
>
> **Our revision:**
> * We performed both qualitative (**Figure 3, Page 7**) and quantitative (**Table 1, Page 6**) evaluation on more challenging real-world tasks (**task setting is illustrated in Figure 10, Page 16**), involving unseen objects and unseen background.
> * We provided evaluation on real-world tasks involving occlusion (**Figure 16, Page 31**).
>
> ---
>
> ### Reviewer 1NeT
> **Concerns:**
> * Lack of quantitative analysis on the real-world tasks.
> * Need to evaluate the benefit of self-discriminated filtering on more tasks.
> * Lack of quantitative evaluation of 'Strong dynamics model' across all environments.
>
> **Our revision:**
> * We provided a quantitative analysis on the real-world tasks in terms of both dynamics accuracy (**Figure 5&7, Page 7&8**) and planning success rates (**Table 1, Page 6**).
> * We added an extra ablation study of self-discriminated filtering on real-world tasks (**Figure 7, Page 8**).
> * We provided a quantitative evaluation of 'Strong dynamics model' across all environments (**Figure 5, Page 7**).
>
> ---
>
> ### Reviewer rs1q
> **After discussion with Reviewer rs1q, his concerns were all addressed and he had raised his score to 6!**
>
> ---
>
> ### Remark
> Overall, all reviewers recognized that our method is novel and addresses an important pain point (hallucination) in high-level multimodal planning. Most concerns focused on experimental completeness, and we have followed all reviewer suggestions and added all experiments.
>
> We believe the revised paper now provides comprehensive evidence, resolves all reviewer concerns, and fairly reflects the contribution of our work. We sincerely hope the AC will consider our revisions fairly. Thank you very much.

---

### Meta-Review · Area_Chair_4cif · 2025-12-18

**Summary:**

The paper proposes Uni-Plan, a high-level planning model built on a unified multimodal model BAGEL. Uni-Plan simultaneously functions as a policy, dynamics model, self-discriminator and value function. Experiments on long-horizon planning tasks demonstrate the effectiveness of the approach.

The initial scores from the four reviewers are 2, 4, 4, and 4. The major concerns raised include:
1) missing comparisons with prior planning methods and world-model-based approaches (Lo2k, rs1q);
2) lack of statistical rigor in the experimental results (Lo2k);
3) evaluation on relatively simple tasks with full state observability and limited long-horizon difficulty (VZiD, 1NeT, rs1q);
4) insufficient generalization studies and absence of challenging real-robot experiments (VZiD);
5) lack of comprehensive quantitative analysis across all environments (1NeT).

The rebuttal partially addresses some of these concerns. However, points 1, 3, 5 remain insufficiently resolved in the AC’s assessment.
The evaluated tasks are not sufficiently challenging and assume full initial observability. While the authors argue that performing open-loop planning in the first place is an advantage, this claim is not well supported, especially for dynamic environments where closed-loop planning is more effective.
Additionally, the ablation studies are limited. For example, it is unclear how the proposed unified model compares against architectures that use separate modules for policy, dynamics, and value estimation, or against established planning methods that leverage explicit world models [1].
[1] Wu Y, Tian T, Swamy G, et al. From Foresight to Forethought: VLM-In-the-Loop Policy Steering via Latent Alignment. RSS 2025.

Therefore, the AC recommends rejecting the paper.

**Reviewer Concerns:**

See the summary.

**Reviewer Scores:**

Given the remaining concerns, the reviewers’ scores are expected to remain the same.

---

### Decision · Program_Chairs · 2026-01-26

Reject